# Reexamination of N-terminal domains of syntaxin-1 in vesicle fusion from central murine synapses

Gülçin Vardar[1,2]*, Andrea Salazar-Lázaro[1,2], Marisa Brockmann[1,2], Marion Weber-Boyvat[1,2], Sina Zobel[1,2], Victor Wumbor-Apin Kumbol[3], Thorsten Trimbuch[1,2], Christian Rosenmund[1,2]*

[1]Universität Berlin, Humboldt-Universität zu Berlin, Berlin, Germany; [2]Berlin Institute of Health, Berlin, Germany; [3]Einstein Center for Neurosciences Berlin, Berlin, Germany

*For correspondence:
gulcinv@gmail.com (GV);
christian.rosenmund@charite.
de (CR)

Competing interest: The authors declare that no competing interests exist.

**Abstract** Syntaxin-1 (STX1) and Munc18-1 are two requisite components of synaptic vesicular release machinery, so much so synaptic transmission cannot proceed in their absence. They form a tight complex through two major binding modes: through STX1's N-peptide and through STX1's closed conformation driven by its $H_{abc}$- domain. However, physiological roles of these two reportedly different binding modes in synapses are still controversial. Here we characterized the roles of STX1's N-peptide, $H_{abc}$-domain, and open conformation with and without N-peptide deletion using our STX1-null mouse model system and exogenous reintroduction of STX1A mutants. We show, on the contrary to the general view, that the $H_{abc}$-domain is absolutely required and N-peptide is dispensable for synaptic transmission. However, STX1A's N-peptide plays a regulatory role, particularly in the $Ca^{2+}$-sensitivity and the short-term plasticity of vesicular release, whereas STX1's open conformation governs the vesicle fusogenicity. Strikingly, we also show neurotransmitter release still proceeds when the two interaction modes between STX1A and Munc18-1 are presumably intervened, necessitating a refinement of the conceptualization of STX1A–Munc18-1 interaction.

## Introduction

The synaptic vesicle (SV) fusion is the fundamental process in synaptic transmission, and it is catalyzed by the merger of plasma and vesicular membranes by the neuronal SNAREs syntaxin-1 (STX1 collectively refers to STX1A and STX1B throughout this study), synaptobrevin-2 (Syb-2), and SNAP25 (*Rizo and Sudhof, 2012*; *Rizo and Xu, 2015*; *Baker and Hughson, 2016*). STX1 is the most important neuronal SNARE because not only synaptic transmission grinds to a halt in its absence, but also neurons cannot survive (*Vardar et al., 2016*). Compared to the other SNAREs, it also has a unique structure with its regulatory region composed of a bulky three helical $H_{abc}$-domain and a short N-peptide preceding its SNARE motif (*Figure 1A*; *Fernandez et al., 1998*).

Besides its interaction with the other SNAREs, STX1 also binds to its cognate SM protein Munc18-1 forming a tight binary complex with an affinity in the nanomolar range (*Pevsner et al., 1994*; *Burkhardt et al., 2008*). Munc18-1, which is an assistor of SNARE-mediated vesicular release, is an equally important protein as its absence also leads to inhibition of synaptic transmission (*Verhage et al., 2000*). Two major modes for STX1 binding to Munc18-1 have been defined: one through its N-peptide, the other through its closed conformation driven by the intramolecular interaction between its $H_{abc}$- and SNARE domains (*Dulubova et al., 1999*; *Misura et al., 2000*). However, several issues regarding these reportedly different binding modes of STX1 to Munc18-1 are still subjects of dispute.

It is evident that STX1's $H_{abc}$-domain is required for proper folding of STX1 and for proper co-recruitment of STX1–Munc18-1 complex to the active zone (AZ) (**Han et al., 2009**; **Meijer et al., 2012**; **Vardar et al., 2020**; **Zhou et al., 2013**), yet it has been deemed to play a secondary role in synaptic transmission, to the point that it is dispensable for vesicle fusion *per se* (**Rathore et al., 2010**; **Shen et al., 2010**; **Meijer et al., 2012**; **Zhou et al., 2013**). However, an increasing number of mutations discovered in the $H_{abc}$-domain of STX1B in patients with epilepsy (**Schubert et al., 2014**; **Wolking et al., 2019**; **Vardar et al., 2020**) points to greater importance for this region in neurotransmitter release.

The physiological significance of Munc18-1 binding to STX1's N-peptide is less clear, even though the general view leans towards its indispensability for synaptic transmission. Firstly, the STX1 N-peptide does not majorly contribute to its overall affinity for Munc18-1 (**Burkhardt et al., 2008**; **Christie et al., 2012**; **Colbert et al., 2013**), yet liposome fusion cannot proceed without the N-peptide in reconstitution experiments (**Shen et al., 2007**; **Rathore et al., 2010**; **Shen et al., 2010**). On the other hand, interfering with STX1-N-peptide–Munc18-1 interaction by mutations either on STX1 (**Zhou et al., 2013**; **Park et al., 2016**) or on Munc18-1 (**Khvotchev et al., 2007**; **Shen et al., 2007**; **Han et al., 2009**; **Meijer et al., 2012**) in synapses in diverse model systems disclosed either its essentiality or its dispensability. Thus, a collective consensus as to what function the binding of STX1's highly conserved N-peptide to Munc18-1 plays in synaptic transmission has not been reached.

So far, the physiological roles of STX1's N-peptide, $H_{abc}$-domain, and open-closed conformation were not assessed in central synapses completely devoid of STX1. Rather, studies have been conducted either in synapses with normal STX1 expression but mutant Munc18-1 (**Khvotchev et al., 2007**; **Meijer et al., 2012**; **Shen et al., 2018**) or in synapses with only severely reduced expression of STX1 (**Zhou et al., 2013**). Furthermore, in vitro studies do not contain the full panel of native synaptic proteins and mostly do not use full-length STX1 (**Shen et al., 2007**; **Rathore et al., 2010**; **Shen et al., 2010**). Therefore, we addressed the contribution of different domains of STX1 to neurotransmission using our STX1-null mouse model system and exogenous reintroduction of STX1A mutants either lacking N-peptide or the $H_{abc}$-domain, or STX1 mutants forced into the open conformation ($LE_{Open}$ mutation) with or without an N-peptide deletion. We show that the $H_{abc}$-domain is absolutely required for STX1's stability and/or expression and thus neurotransmitter release. Furthermore, in contrast to the general view, we find that N-peptide is not indispensable for synaptic transmission; however, we propose that STX1's N-peptide plays a regulatory role, particularly in the $Ca^{2+}$-sensitivity of vesicular release and generally in vesicle fusion, which is only unmasked by STX1's open conformation.

## Results

### STX1's $H_{abc}$-domain is essential and N-peptide is dispensable for neurotransmitter release

Vesicle fusion does not occur in the absence of STX1 (**Vardar et al., 2016**) providing a null background in terms of neurotransmitter release. Thus, we used STX1A constitutive, STX1B conditional knockout (STX1-null) mouse neurons and lentiviral expression of different STX1 mutants in conjunction with *Cre* recombinase (**Vardar et al., 2016**; **Vardar et al., 2020**) to study the structure–function relationship of STX1 domains. With the focus on the effects of different Munc18-1 binding modes, we expressed STX1A mutants either with the deletion of the N-peptide ($STX1A^{\Delta N2-9}$) or the $H_{abc}$-domain ($\Delta29-144$; $STX1A^{\Delta Habc}$) or with the introduction of well-described $LE_{Open}$ (L165A, E166A; $STX1A^{LEOpen}$) mutation (**Figure 1A**).

Firstly, we utilized immunocytochemistry in high-density hippocampal neuronal culture to quantify the exogenous expression of $STX1A^{\Delta N2-9}$, $STX1A^{\Delta Habc}$, and $STX1A^{LEOpen}$ at presynaptic compartments as defined by Bassoon-positive puncta and normalized fluorescence signals to the signals caused by expression of $STX1A^{WT}$, all in STX1-null neurons. As expected from previous studies (**Meijer et al., 2012**; **Zhou et al., 2013**), deletion of the N-peptide had no significant effect on STX1A expression compared to $STX1A^{WT}$, whereas $STX1A^{\Delta Habc}$ did not produce a measurable signal (**Figure 1B and C**). Loss of STX1 leads to a severe reduction in Munc18-1 expression, which can be rescued by the expression of either STX1A or STX1B (**Zhou et al., 2013**; **Vardar et al., 2016**; **Vardar et al., 2020**). Consistent with the expected relative binding states of $STX1A^{\Delta N2-9}$ and $STX1A^{\Delta Habc}$ to Munc18-1 (**Burkhardt et al., 2008**), N-peptide deletion did not cause a significant change in Munc18-1 expression at

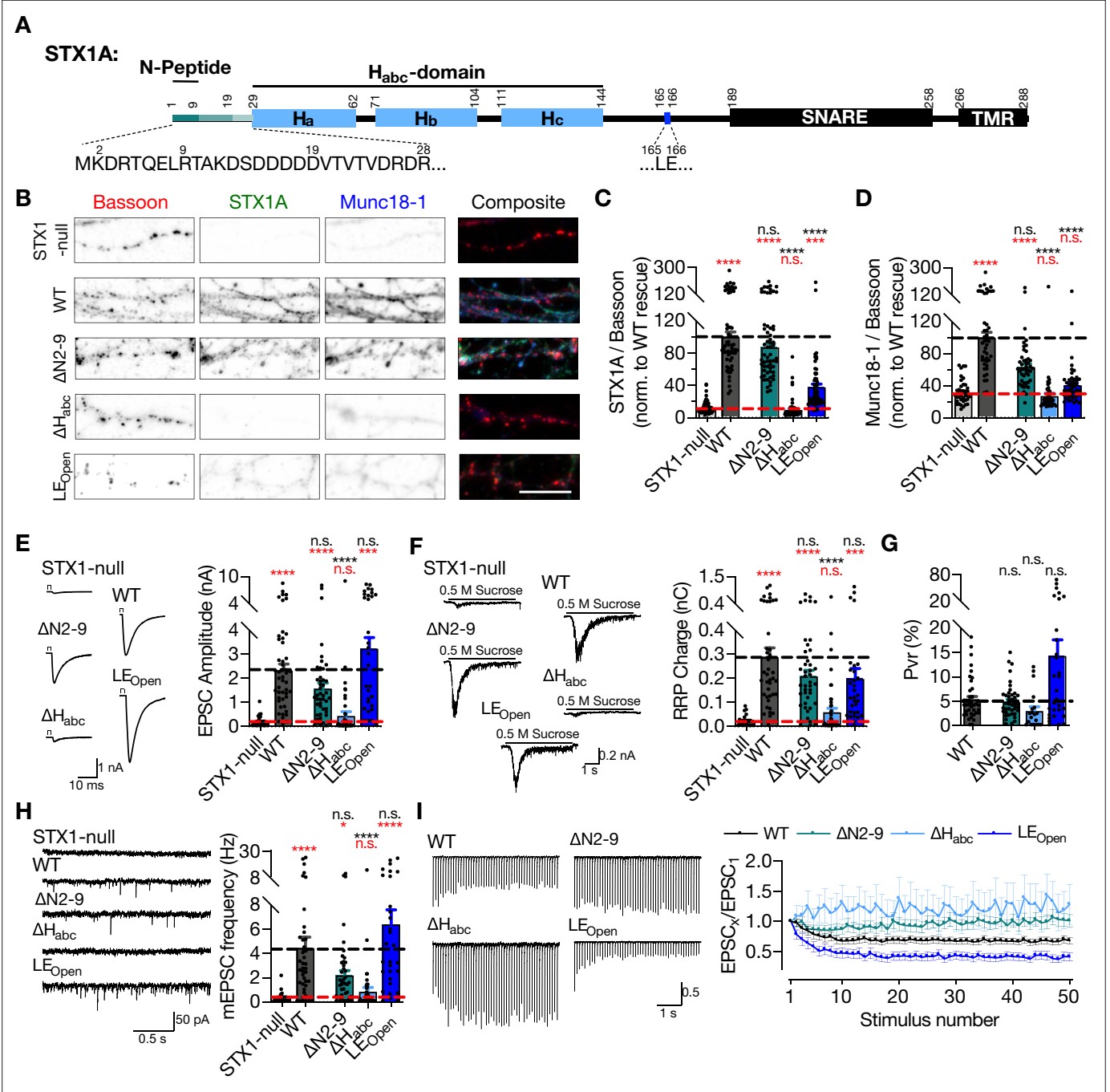

**Figure 1.** STX1A's $H_{abc}$-domain is essential and N-peptide is dispensable for neurotransmitter release. (**A**) Domain structure of STX1A. The protein consists of a short N-peptide (aa 1–9 or 1–28), $H_{abc}$ domain (aa 29–144) formed by three helices, $H_a$, $H_b$, and $H_c$, followed by the H3 helix (aa 189–259; SNARE domain) and a transmembrane region (aa 266–288; TMR). (**B**) Example images of immunofluorescence labeling for Bassoon, STX1A, and Munc18-1 shown as red, green, and blue, respectively, in the corresponding composite pseudocolored images obtained from high-density cultures of STX1-null hippocampal neurons either not rescued or rescued with STX1A$^{WT}$, or STX1A$^{Δ2-9}$; STX1A$^{LEOpen}$; or STX1A$^{ΔHabc}$. Scale bar: 10 μm (**C, D**) Quantification of the immunofluorescence intensity of STX1A and Munc18-1 as normalized to the immunofluorescence intensity of Bassoon in the same ROIs as shown in (**B**). The values were then normalized to the values obtained from STX1A$^{WT}$ neurons. (**E**) Example traces (left) and quantification of the amplitude (right) of EPSCs obtained from hippocampal autaptic STX1-null neurons either not rescued or rescued with STX1A$^{WT}$, STX1B$^{Δ2-9}$, STX1A$^{LEOpen}$, or STX1A$^{ΔHabc}$. (**F**) Example traces (left) and quantification of the charge transfer (right) of 500 mM sucrose-elicited readily releasable pools (RRPs) obtained from the same neurons as in (**E**). (**G**) Quantification of probability of vesicular release (Pvr) determined as the percentage of the RRP released upon one AP. (**H**) Example traces (left) and quantification of the frequency (right) of mEPSCs recorded at –70 mV. (**I**) Example traces (left) and quantification (right) of short-term plasticity (STP) determined by high-frequency stimulation at 10 Hz and normalized to the EPSC$_1$ from the same

*Figure 1 continued on next page*

*Figure 1 continued*

neuron. Data information: the artifacts are blanked in example traces in (**D**) and (**H**). The example traces in (**G**) were filtered at 1 kHz. In (**C–H**), data points represent single observations, the bars represent the mean ± SEM. In (**I**), data points represent mean ± SEM. Red and black annotations (stars and n.s.) on the graphs show the significance comparisons to STX1-null and to STX1A$^{WT}$ rescue, respectively (nonparametric Kruskal–Wallis test followed by Dunn's *post hoc* test, *p≤0.05, ***p≤0.001, ****p≤0.0001). Two-way ANOVA was applied for data in (**I**). The numerical values are summarized in *Figure 1—source data 1*.

The online version of this article includes the following source data and figure supplement(s) for figure 1:

**Source data 1.** Quantification of the STX1A$^{WT}$ and mutant STX1A expression induced by lentiviral transduction of STX1-null neurons and the consequent neurotransmitter release properties.

**Figure supplement 1.** STX1A$^{ΔHabc}$ expression cannot be detected.

**Figure supplement 1—source data 1.** Quantification of the expression of FLAG-tagged WT and mutant STX1A.

Bassoon positive puncta, whereas the H$_{abc}$-domain deletion was unable to rescue Munc18-1 levels back to WT-like levels (*Figure 1B and D*). Rendering STX1B constitutively open by LE$_{Open}$ mutation is also known to decrease STX1B as well as Munc18-1 levels (*Gerber et al., 2008*) and the expression of STX1A$^{LEOpen}$ was severely low and inefficient to rescue Munc18-1 levels (Figure 1B–D).

To assess how the manipulation of the different Munc18-1 binding domains of STX1A affect the release of presynaptic vesicles, we measured Ca$^{2+}$-triggered and spontaneous vesicle fusion, vesicle priming, and short-term plasticity (STP) in autaptic hippocampal neurons using electrophysiology as described previously (*Vardar et al., 2016*; *Vardar et al., 2020*). Compared to STX1A$^{WT}$ neurons, STX1A$^{LEOpen}$ neurons exhibited a trend towards a 40 % increase in EPSC (*Figure 1E*) and towards a 30 % decrease in hypertonic-sucrose measured readily releasable pool (RRP) (*Figure 1F*), trending towards an approximately threefold increase in probability of vesicular release (Pvr) (*Figure 1G*). The increase in Pvr, though not significant, was also evident in the observed enhancement of short-term depression (*Figure 1I*) as well as in the trend towards increased mEPSC frequency (*Figure 1H*). These findings are consistent with the previous studies on the LE$_{Open}$ mutation on STX1A or STX1B (*Gerber et al., 2008*; *Zhou et al., 2013*).

Surprisingly, loss of N-peptide of STX1A showed only a trend towards 30 % decrease in Ca$^{2+}$-evoked vesicular release (*Figure 1E*), but not its full arrest. Similarly, RRP and spontaneous neurotransmission, which is assessed by the frequency of single-vesicle release events, were not completely inhibited by N-peptide deletion, but only trended towards a decrease by 30 and 50%, respectively, (*Figure 1F and H*). Proportionally similar trends in the reduction of both EPSC and RRP resulted in comparable Pvr between STX1A$^{ΔN2-9}$ and STX1A$^{WT}$ neurons (*Figure 1G*). Despite the lack of a net difference in the Pvr, however, STX1A$^{ΔN2-9}$ neurons exhibited an altered STP in response to the 10 Hz stimulation, with no depression to latter stimuli (*Figure 1I*).

Previous studies have suggested that the H$_{abc}$-domain of STX1A and particularly its interaction with Munc18-1 is dispensable for vesicle fusion both in vitro and in vivo (*Rathore et al., 2010*; *Shen et al., 2010*; *Meijer et al., 2012*; *Zhou et al., 2013*). However, our analysis of the neurotransmission properties of the STX1A$^{ΔHabc}$ neurons in comparison to the STX1A$^{WT}$ neurons showed that STX1A$^{ΔHabc}$ was incapable of rescuing neurotransmitter release as it produced no detectable EPSC, RRP, or mEPSC; a phenotype similar to the STX1-null neurons (Figure 1E–G).

## STX1 H$_{abc}$-domain is indispensable for neuronal viability and the organization of synaptic ultrastructure

STX1 has also an obligatory function in neuronal maintenance and complete loss of both STX1A and STX1B leads to neuronal death (*Vardar et al., 2016*). To address the overall functionality of STX1A, we assessed the survivability of the high-density cultured STX1-null neurons expressing STX1A$^{ΔHabc}$ and determined the cell number at different time intervals starting at DIV 8 (*Figure 2A–C*), at which time point all the groups had an average of ~40 neurons per mm$^2$ (*Figure 2B*). Then we calculated the ratio of the cell number at DIV 15, 22, and 29 to the cell number at DIV 8 as a read-out for neuronal viability. STX1-null neurons showed a dramatic loss between DIV 8 and DIV 15 (*Figure 2C*) as reported before (*Vardar et al., 2016*). Even though at DIV 15 the number of surviving STX1A$^{ΔHabc}$ neurons was slightly but significantly higher compared to that in STX1-null group, eventually STX1A$^{ΔHabc}$ failed to rescue neuronal survival as by DIV 22 almost all STX1A$^{ΔHabc}$ neurons were dead (*Figure 2C*).

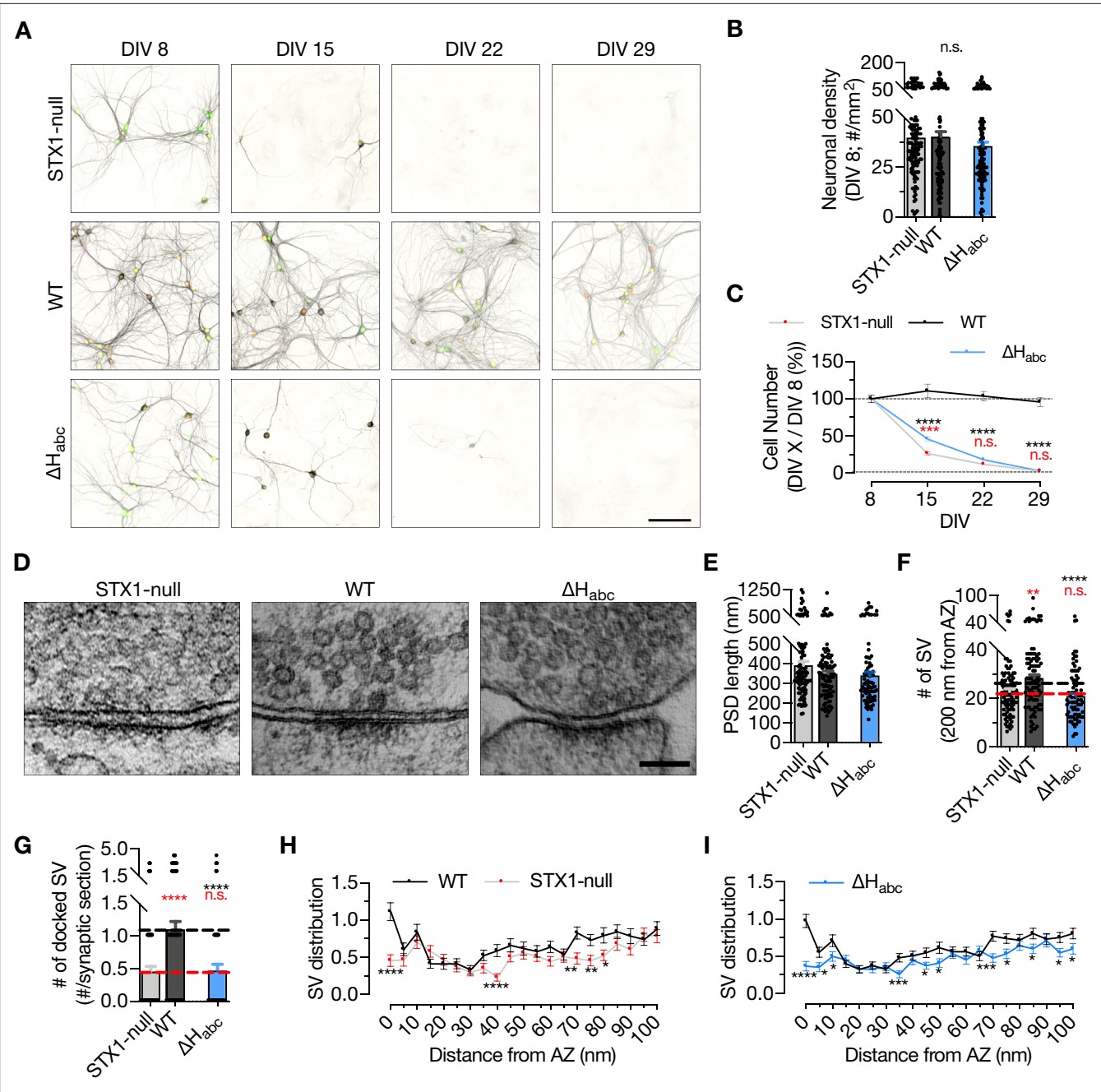

**Figure 2.** STX1's $H_{abc}$-domain is essential for the overall function of STX1A. (**A**) Example images of high-density cultures of STX1-null, STX1A[WT], and STX1A[ΔHabc] hippocampal neurons at DIV 8, 15, 22, and 29 represented with immunofluorescent labeling of microtubule associated protein 2 (MAP2) . Red and green nuclei serve as a marker for NLS-RFP-P2A-*Cre* recombinase expression and for NLS-GFP-P2A-STX1A (either WT or mutants), respectively. Scale bar: 50 µm. (**B**) Quantification of neuronal density at DIV 8. (**C**) Quantification of the percentage of the surviving neurons at DIV 8, 15, 22, and 29 as normalized to the neuronal density at DIV 8 in the same well. (**D**) Example high-pressure freezing fixation combined with electron microscopy (HPF-EM) images of nerve terminals from high-density cultures of STX1-null hippocampal neurons either not rescued or rescued with STX1A[WT] or STX1A[ΔHabc]. (**E–G**) Quantification of active zone (AZ) length, number of synaptic vesicles (SVs) within 200 nm distance from AZ, and number of docked SVs. (**H, I**) SV distribution of STX1-null and STX1A[ΔHabc] neurons compared to that of STX1A[WT] neurons. Data information: in (**B, E–G**), data points represent single observations, the bars represent the mean ± SEM. In (**C, H, I**), data points represent the mean ± SEM. Red and black annotations (stars and n.s.) on the graphs show the significance comparisons to STX1-null and to STX1A[WT] neurons, respectively (nonparametric Kruskal–Wallis test followed by Dunn's *post hoc* test, *p≤0.05, **p≤0.01, ***p≤0.001, ****p≤0.0001). The numerical values are summarized in *Figure 2—source data 1*.

The online version of this article includes the following source data for figure 2:

**Source data 1.** Quantification of the neuronal density at different time intervals and quantification of ultrastructural synaptic properties in high density cultures of STX1-null, STX1A[WT], and STX1A[ΔHabc] neurons.

Furthermore, we analyzed vesicle docking by morphological assessment of synaptic ultrastructure to determine whether STX1A$^{\Delta Habc}$ expression could reverse the impairment in the vesicle docking observed in STX1-null neurons (*Vardar et al., 2016*). To circumvent the reduction in cell number and the synapse number thereof, we transduced the neurons at DIV 2–3 to postpone the cell death as previously shown (*Vardar et al., 2016*) and analyzed the synapses using high-pressure freezing fixation (DIV 14–16) combined with electron microscopy (HPF-EM; *Figure 2D–I*). Firstly, we observed no difference in the postsynaptic density (PSD) length, which is an indirect measurement of the opposing AZ length, among STX1-null, STX1A$^{WT}$, and STX1A$^{\Delta Habc}$ neurons (*Figure 2E*). On the other hand, the total SV number within 200 nm from the AZ was significantly reduced in STX1A$^{\Delta Habc}$ neurons compared to that in STX1A$^{WT}$ neurons (*Figure 2F*). STX1A$^{\Delta Habc}$ also did not restore vesicle docking, which remained at ~50 % of the STX1A$^{WT}$ neurons (*Figure 2G*). Similarly, the SV distribution within 100 nm of the AZ were comparable between STX1-null and STX1A$^{\Delta Habc}$ neurons, with both significantly altered number of SVs compared to the STX1$^{WT}$ neurons, especially in the 15, 40, and 100 nm range from AZ (*Figure 2H and I*). This suggests a general alteration of the synaptic organization even though the length of AZs was unaltered.

Based on the lack of immunofluorescent signal (*Figure 1C*) together with the lack of any rescue activity in any release parameters (*Figures 1E–I, 2G and I*) and neuronal survivability for STX1A$^{\Delta Habc}$ (*Figure 2C*), we again examined the expression level of STX1A$^{\Delta Habc}$ in comparison with STX1A$^{WT}$, this time using constructs with a C-terminal FLAG tag (*Figure 1—figure supplement 1*). C-terminal FLAG tag did not reveal significant changes in the expression of STX1A$^{WT}$ (*Figure 1—figure supplement 1*). We then measured the immunofluorescent signal using a FLAG antibody in the neurons expressing FLAG-tagged STX1A$^{WT}$, STX1A$^{\Delta N2-9}$, STX1A$^{LEOpen}$, or STX1A$^{\Delta Habc}$, all of which showed similar levels of reduction in the expression as compared to the non-tagged constructs (*Figure 1—figure supplement 1*), suggesting that the lack of immunofluorescent signal in STX1A$^{\Delta Habc}$ (*Figure 1C*) is not due to a loss of antibody binding epitope, but rather due to the low level of protein.

## Deletion of the entire N-terminal stretch does not impair neurotransmitter release

It is striking that deletion of the 2–9 amino acids (aa), namely the N-peptide, of STX1A revealed no significant phenotype in synaptic transmission from central synapses (*Figure 1*), even though this domain has been designated as a crucial factor for neurotransmitter release. Though the aa 2–9 has been defined as the residues binding to the outer surface of Munc18-1 (*Hu et al., 2007*; *Burkhardt et al., 2008*), the whole 2–28 aa stretch manifests an unstructured nature in NMR studies (*Misura et al., 2000*), suggesting a potential involvement in protein–protein interactions. Thus, we extended our analysis of the function of N-peptide by constructing STX1A with longer deletions in the N-terminus (STX1A$^{\Delta N2-19}$ and STX1A$^{\Delta N2-28}$) and probed the effects of these mutants on synaptic transmission.

Compared to the exogenous expression of STX1A$^{WT}$, deletion of 19 or 28 aa from the N-terminus reduced the expression of STX1A to ~60 % (*Figure 3A–B*), suggesting a modulatory effect of the unstructured N-terminal domain on STX1's expression or stability. However, neither the reduction in STX1A expression nor loss of the putative Munc18-1 binding domain influenced the Munc18-1 levels, which was effectively rescued back to WT-like levels (*Figure 3A and C*).

Strikingly, similar to the deletion of the N-peptide, neither deletion of 2–19 aa nor 2–28 aa led to full inhibition of vesicle fusion nor of vesicle priming, but only a graded trend towards a decrease by 20–30% (*Figure 3D and E*). STX1A$^{WT}$ and STX1A$^{\Delta N2-9}$ neurons had an average EPSC of ~6 nA and an average RRP of ~0.5 nC, while STX1A$^{\Delta N2-19}$ and STX1A$^{\Delta N2-28}$ had an average EPSC of ~4 nA and an average RRP of ~0.4 nC (*Figure 3D and E*). A trend towards a reduction in release was also expressed in Pvr, such that STX1A$^{\Delta N2-19}$ and STX1A$^{\Delta N2-28}$ neurons manifested Pvr of ~6 %, whereas STX1A$^{WT}$ and STX1A$^{\Delta N2-9}$ neurons released with a Pvr of ~8 % and ~7%, respectively (*Figure 3F*). As another measure of Pvr, we induced paired action potentials (APs) at 40 Hz and observed no difference in paired-pulse ratio (PPR) of EPSCs between STX1A$^{WT}$ and STX1A$^{\Delta N}$ neurons (*Figure 3G*). Similarly, spontaneous release inclined to be impaired by 30–45% but not significantly, remaining at around 3–4 Hz compared to ~6 Hz of STX1A$^{WT}$ (*Figure 3H*). A similar level of reduction both in mEPSC frequency and RRP size recorded from STX1A$^{\Delta N2-19}$ and STX1A$^{\Delta N2-28}$ neurons led to no difference in spontaneous vesicle fusion rate compared to that recorded from STX1A$^{WT}$ neurons (*Figure 3I*).

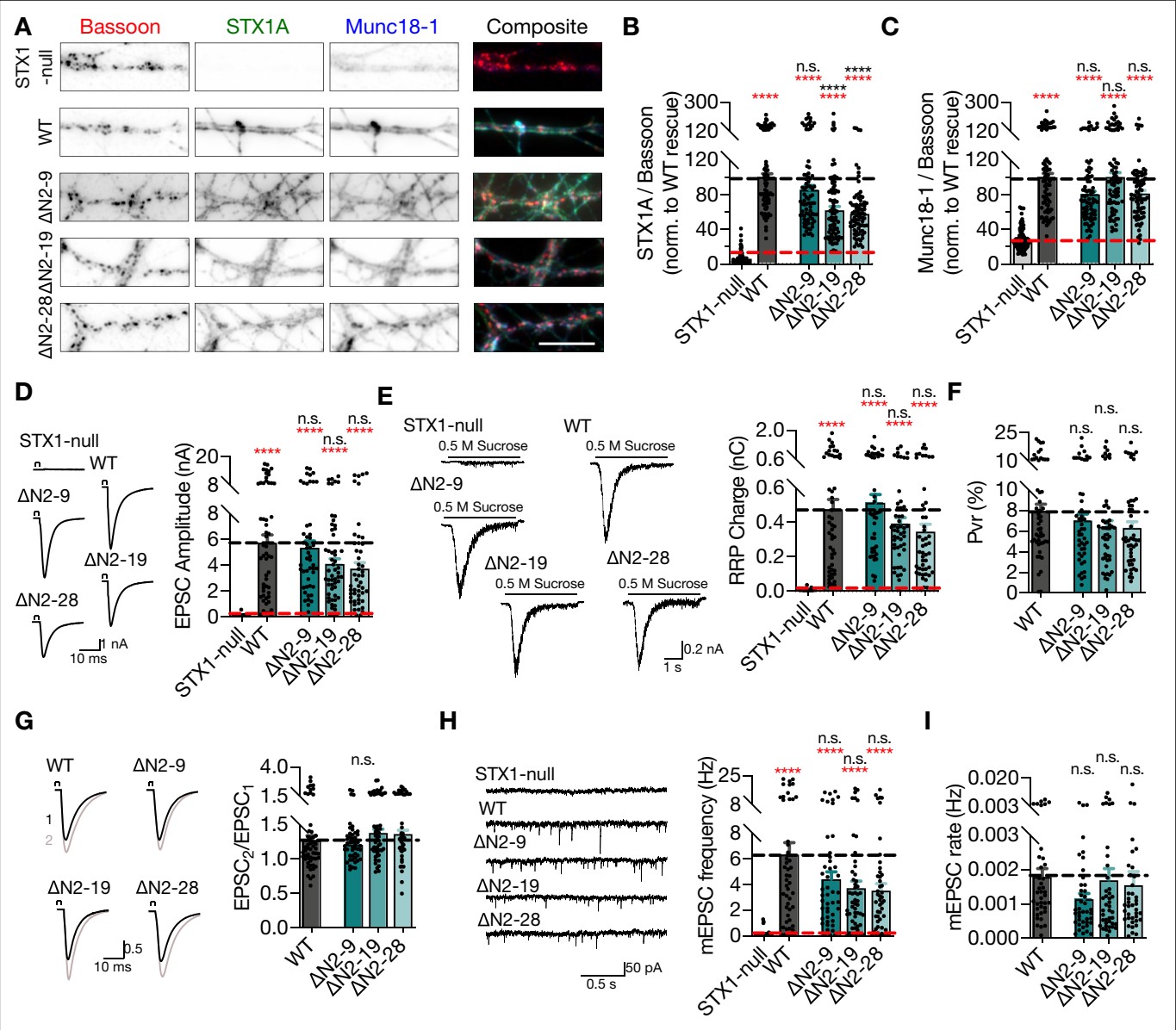

**Figure 3.** Deletion of the entire N-terminal stretch does not impair neurotransmitter release. (**A**) Example images of immunofluorescence labeling for Bassoon, STX1A, and Munc18-1 shown as red, green, and blue, respectively, in the corresponding composite pseudocolored images obtained from high-density cultures of STX1-null hippocampal neurons either not rescued or rescued with STX1A$^{WT}$, STX1A$^{\Delta2-9}$, STX1A$^{\Delta2-19}$, or STX1A$^{\Delta2-28}$. Scale bar: 10 μm. (**B, C**) Quantification of the immunofluorescence intensity of STX1A and Munc18-1 as normalized to the immunofluorescence intensity of Bassoon in the same ROIs as shown in (**A**). The values were then normalized to the values obtained from STX1A$^{WT}$ neurons. (**D**) Example traces (left) and quantification of the amplitude (right) of EPSCs obtained from hippocampal autaptic STX1-null neurons either not rescued or rescued with STX1A$^{WT}$, STX1A$^{\Delta2-9}$, STX1A$^{\Delta2-19}$, or STX1A$^{\Delta2-28}$. (**E**) Example traces (left) and quantification of the charge transfer (right) of sucrose-elicited readily releasable pools (RRPs) obtained from the same neurons as in (**D**). (**F**) Quantification of probability of vesicular release (Pvr) determined as the percentage of the RRP released upon one action potential (AP). (**G**) Example traces (left) and quantification (right) of paired-pulse ratio (PPR) measured at 40 Hz. The artifacts are blanked in the example traces. (**H**) Example traces (left) and quantification of the frequency (right) of mEPSCs. The example traces were filtered at 1 kHz. (**I**) Quantification of mEPSC rate as spontaneous release of one unit of RRP. Data information: the artifacts are blanked in example traces in (**D**) and (**G**). The example traces in (**H**) were filtered at 1 kHz. In (**B–I**), data points represent single observations, the bars represent the mean ± SEM. Red and black annotations (stars and n.s.) on the graphs show the significance comparisons to STX1-null and to STX1A$^{WT}$ neurons, respectively (nonparametric Kruskal–Wallis test followed by Dunn's *post hoc* test, ****p≤0.0001). The numerical values are summarized in *Figure 3—source data 1*.

The online version of this article includes the following source data and figure supplement(s) for figure 3:

**Source data 1.** Quantification of the lentiviral expression of STX1A$^{WT}$ and STX1A$^{\Delta N}$ mutants in STX1-null neurons and the consequent neurotransmitter release properties.

*Figure 3 continued*

**Figure supplement 1.** Phosphorylation of S14 of STX1 has no effect on synaptic transmission.

**Figure supplement 1—source data 1.** Quantification of lentiviral expression of phosphonull and phosphomimetic STX1A mutants and the consequent neurotransmitter release properties.

**Figure supplement 2.** Modifications of STX1's N-peptide either by deletions or phosphorylation does not compromise neuronal viability.

**Figure supplement 2—source data 1.** Quantification of neuronal density of STX1A$^{\Delta N}$ and phosphorylation mutant neurons in comparison to STX1A$^{WT}$ and STX1-null neurons at DIV 8 and DIV 29.

Apart from STX1A's first nine aa, the STX1-N-peptide–Munc18-1 interaction is also proposed to be regulated by the phosphorylation of STX1's S14 residue by CKII (*Rickman and Duncan, 2010*). To test whether the phosphorylation of S14 affects Munc18-1 trafficking and neurotransmitter release, we generated phosphonull (S14A) and phosphomimetic (S14E) STX1A mutants. We again measured the STX1A and Munc18-1 levels at synapses, which revealed no impact of the phosphorylation status of S14 on either STX1A or Munc18-1 levels (*Figure 3—figure supplement 1*), consistent with the finding that S14A causes only a minor decrease in the affinity of STX1A to Munc18-1 (*Burkhardt et al., 2008*). As a direct function of STX1A S14 phosphorylation on vesicular release from neurons or neuroendocrine cells has been also proposed (*Rickman and Duncan, 2010*; *Shi et al., 2020*), we tested whether it would also influence the fusion of presynaptic vesicles. Both STX1A$^{S14A}$ and STX1A$^{S14E}$ efficiently restored all the release parameters to WT-like levels in STX1-null neurons (*Figure 3—figure supplement 1*), which suggests that the modulation of the STX1A N-peptide–Munc18-1 interaction by S14 phosphorylation does not alter its function in neurotransmitter release from central synapses. Neither N-peptide deletion nor phosphorylation modulation mutants compromised the neuronal survival (*Figure 3—figure supplement 2*).

## 'Opening' of STX1A in combination with the deletion of its entire N-terminal stretch does not impair neurotransmitter release

Munc18-1 binding to the N-peptide or to the closed conformation of STX1 constitutes the two well-defined interaction modes between these proteins, yet neither mutation causes a major deficit in synaptic release (*Figures 1 and 3*). However, Munc18-1 interacts with STX1A$^{WT}$ through multiple interaction points including the SNARE motif of STX1A (*Misura et al., 2000*; *Burkhardt et al., 2008*; *Liang et al., 2013*). To test whether or not the modulation of both 'closed' and 'N-peptide' binding modes would result in a drastic loss of the STX1A–Munc18-1 binary complex (*Rickman et al., 2007*) and thereby a loss of neurotransmitter release, we constructed STX1A mutants in which the N-peptide is deleted at differing lengths in conjunction with the LE$_{Open}$ mutation. Firstly, we observed that N-peptide deletion in addition to the LE$_{open}$ mutation decreased the STX1A and Munc18-1 levels further than that already caused by LE$_{Open}$ mutation alone (*Figure 4A–C*).

Despite the presumed loss of the two STX1A–Munc18-1 interaction modes, it is remarkable that all LE$_{Open}$-$\Delta$N combination mutants rescued Ca$^{2+}$-evoked neurotransmitter release to almost STX1A$^{WT}$ levels with only a trend towards a reduction by 25–35% (*Figure 4D*). Because STX1A$^{LEOpen}$ neurons showed increased EPSCs – albeit not significant – with an average of ~7 nA compared to ~4 nA of STX1A$^{WT}$, EPSCs recorded from STX1A$^{LEOpen+\Delta N}$ neurons were significantly smaller than that of STX1A$^{LEOpen}$ neurons and remained at ~3 nA (*Figure 4D*). This suggests that the small enhancement of Ca$^{2+}$-evoked release by the presumed open conformation by LE$_{Open}$ mutation was reversed by additional N-peptide deletions (*Figure 4D*). On the other hand, the reduction in RRP observed in neurons that express LE$_{Open}$ mutation was not reverted back to WT-like levels by the addition of N-peptide deletions, but instead was further exaggerated as the RRP size significantly decreased in STX1A$^{LEOpen+\Delta N}$ neurons compared to that in STX1A$^{WT}$ neurons (*Figure 4E*). As a result, increased Pvr, which is the hallmark phenotype of the LE$_{Open}$ mutation (*Gerber et al., 2008*), was reversed back to WT-like levels with only a trend toward a small increase (*Figure 4F*). Increased Pvr in STX1A$^{LEOpen}$-expressing neurons led to decreased PPR when measured at 40 Hz, and N-peptide deletions in STX1A$^{LEOpen}$ reverted PPR back to levels comparable to neurons expressing STX1A$^{WT}$ (*Figure 4G*). Similarly, mEPSC frequency and mEPSC release rate obtained from the STX1A$^{LEOpen+\Delta N}$ mutants were significantly smaller than that of STX1A$^{LEOpen}$ mutant (*Figure 4H and I*).

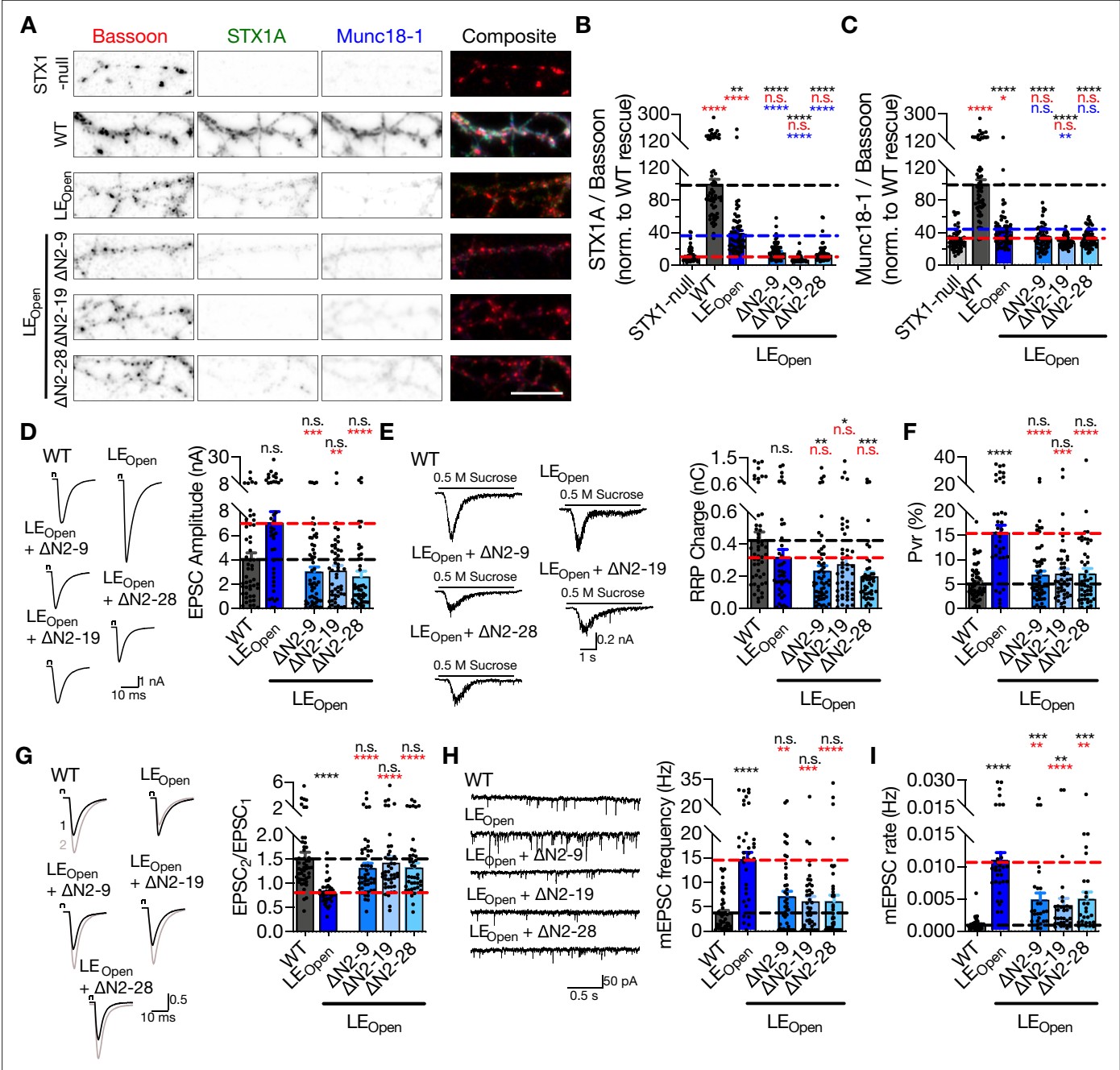

**Figure 4.** 'Opening' of STX1A in combination with the deletion of its entire N-terminal stretch does not impair neurotransmitter release. (**A**) Example images of immunofluorescence labeling for Bassoon, STX1A, and Munc18-1 shown as red, green, and blue, respectively, in the corresponding composite pseudocolored images obtained from high-density cultures of STX1-null hippocampal neurons either not rescued or rescued with STX1A$^{WT}$, STX1A$^{LEOpen}$, STX1A$^{LEOpen + \Delta N2-9}$, STX1A$^{LEOpen + \Delta N2-19}$, or STX1A$^{LEOpen + \Delta N2-28}$. Scale bar: 10 μm (**B, C**) Quantification of the immunofluorescence intensity of STX1A and Munc18-1 as normalized to the immunofluorescence intensity of Bassoon in the same ROIs as shown in (**A**). The values were then normalized to the values obtained from STX1A$^{WT}$ neurons. (**D**) Example traces (left) and quantification of the amplitude (right) of EPSCs obtained from hippocampal autaptic STX1A$^{WT}$, STX1A$^{LEOpen}$, STX1A$^{LEOpen + \Delta N2-9}$, STX1A$^{LEOpen + \Delta N2-19}$, or STX1A$^{LEOpen + \Delta N2-28}$ neurons. (**E**) Example traces (left) and quantification of the charge transfer (right) of sucrose-elicited readily releasable pools (RRPs) obtained from the same neurons as in (**D**). (**F**) Quantification of probability of vesicular release (Pvr) determined as the percentage of the RRP released upon one action potential (AP). (**G**) Example traces (left) and quantification (right) of paired-pulse ratio (PPR) measured at 40 Hz. (**H**) Example traces (left) and quantification of the frequency (right) of mEPSCs. (**I**) Quantification of mEPSC rate as spontaneous release of one unit of RRP. (**I**) Quantification of mEPSC rate as spontaneous release of one unit of RRP.

The online version of this article includes the following source data and figure supplement(s) for figure 4:

**Source data 1.** Quantification of lentiviral expression of STX1A$^{LEOpen}$ and STX1A$^{LEOpen + \Delta N}$ mutants in STX1-null neurons and the consequent

*Figure 4 continued on next page*

*Figure 4 continued*

neurotransmitter release properties.

**Figure supplement 1.** Interruption of both Munc18-1 binding modes of STX1 ultimately leads to neuronal death.

**Figure supplement 1—source data 1.** Quantification of neuronal density of neurons expressing STX1A$^{LEOpen}$ or STX1A$^{LEOpen + \Delta N}$ mutants at different time intervals.

**Figure supplement 2.** Reducing the expression levels of STX1A$^{WT}$ or STX1A$^{LEOpen}$ does not alter their synaptic release properties.

**Figure supplement 2—source data 1.** Quantification of the effects lentiviral downtitration of STX1A$^{WT}$ and STX1A$^{LEOpen}$ on STX1A's and Munc18-1's expression levels and on neurotransmitter release parameters.

**Figure supplement 3.** Exogenous expression of STX1A using 1× volume of lentiviral particles is approximately threefold higher than endogenous STX1A expression.

**Figure supplement 3—source data 1.** Comparison of the lentiviral expression of STX1A with its endogenous expression.

Surprisingly, putative disruption of the two supposedly main interaction points between STX1A and Munc18-1 – by deleting N-peptide in its entirety in LE$_{Open}$ STX1A – ultimately led to neuronal death (*Figure 4—figure supplement 1*) indicative of independence of STX1's functions in neurotransmitter release and neuronal maintenance of one another. However, the onset of cell death was postponed by the expression of STX1A$^{LEOpen+\Delta N}$ mutants compared to that observed in STX1-null neurons as at DIV15 almost all neurons expressing the STX1A$^{LEOpen+\Delta N}$ mutants were still alive (*Figure 4—figure supplement 1*). Because the electrophysiological recordings are mostly conducted at DIV 13–20, the compromised cell viability is unlikely to account for the reduction in neurotransmission in STX1A$^{LEOpen+\Delta N}$ mutants compared to that of in STX1A$^{LEOpen}$ mutant.

A severe reduction in STX1 expression induced by in vitro knock-down (*Arancillo et al., 2013*; *Zhou et al., 2013*) or transgenic knock-in (*Arancillo et al., 2013*) strategies results in a strong impairment in neurotransmitter release. Based upon that, we argued that the reduction of release parameters (*Figure 4D–I*) of STX1A$^{LEOpen}$ by additional N-peptide deletions may be due to decreased expression of STX1A (*Figure 4B*). To test this hypothesis, we down-titrated the viral load from 1 × (~400 × 10$^3$ viral particles per 35 mm well) to 1/12× for STX1A$^{WT}$ and to 1/3× and 1/6× for STX1A$^{LEOpen}$ to reach expression of STX1A at a level comparable to that in STX1A$^{LEOpen+\Delta N2-28}$ neurons (*Figure 4—figure supplement 2*). As our viral constructs include NLS-GFP before the P2A sequence followed by STX1A, nuclear GFP showed a decrease when the amount of virus was reduced (*Figure 4—figure supplement 2*). Immunofluorescent labeling in autaptic neurons revealed that reducing the viral amount was effective in reducing expression levels of either STX1A$^{WT}$ or STX1A$^{LEOpen}$ to the levels comparable to that of STX1A$^{LEOpen+\Delta N2-28}$. However, reducing the exogenous expression level of STX1A$^{WT}$ or STX1A$^{LEOpen}$ down to the level of STX1A$^{LEOpen+\Delta N2-28}$ did not cause a difference in their neurotransmitter release properties (*Figure 4—figure supplement 2*).

We have previously reported that STX1 level becomes a rate-liming factor in neurotransmission when endogenous STX1B expression is knocked-down by shRNA to a level below 20 % on an STX1A-null background (*Arancillo et al., 2013*). Because reducing the lentiviral exogenous expression of STX1A down to ~20 % of the initial experimental conditions did not show any alterations in synaptic release properties (*Figure 4—figure supplement 2*) and thus not reconcile with our previous hypothesis (*Arancillo et al., 2013*), we compared the endogenous STX1A expression in *STX1A$^{+/+}$*; *STX1B$^{+/+}$* neurons to the exogenous expression level in STX1-null neurons transduced with 1× STX1A (*Figure 4—figure supplement 3*). We found that transduction of STX1-null neurons with STX1A using 1× viral volume leads to approximately three fold higher STX1A exogenous expression compared to that of WT neurons. Thus, using 1/12th of the initial viral volume for STX1A transduction leads to an expression level of ~60 % compared to the endogenous level, suggesting that indeed one copy of either *STX1A* or *STX1B* is enough to drive normal synaptic transmission while being insufficient to rescue Munc18-1 levels back to the WT-like levels (*Figure 4—figure supplement 3*). However, please note that our model system does not include STX1B expression and therefore exogenous expression of STX1A in STX1-null neurons by 1× viral volume does not fall within overexpression studies. In summary, our STX1A down-titration experiments show that the reduction in neurotransmitter release properties observed in STX1A$^{LEOpen+\Delta N2-28}$ neurons compared to that of STX1A$^{LEOpen}$ neurons does not stem from lower copy number of STX1A but rather from a functional deficit (*Figure 4—figure supplement 2*).

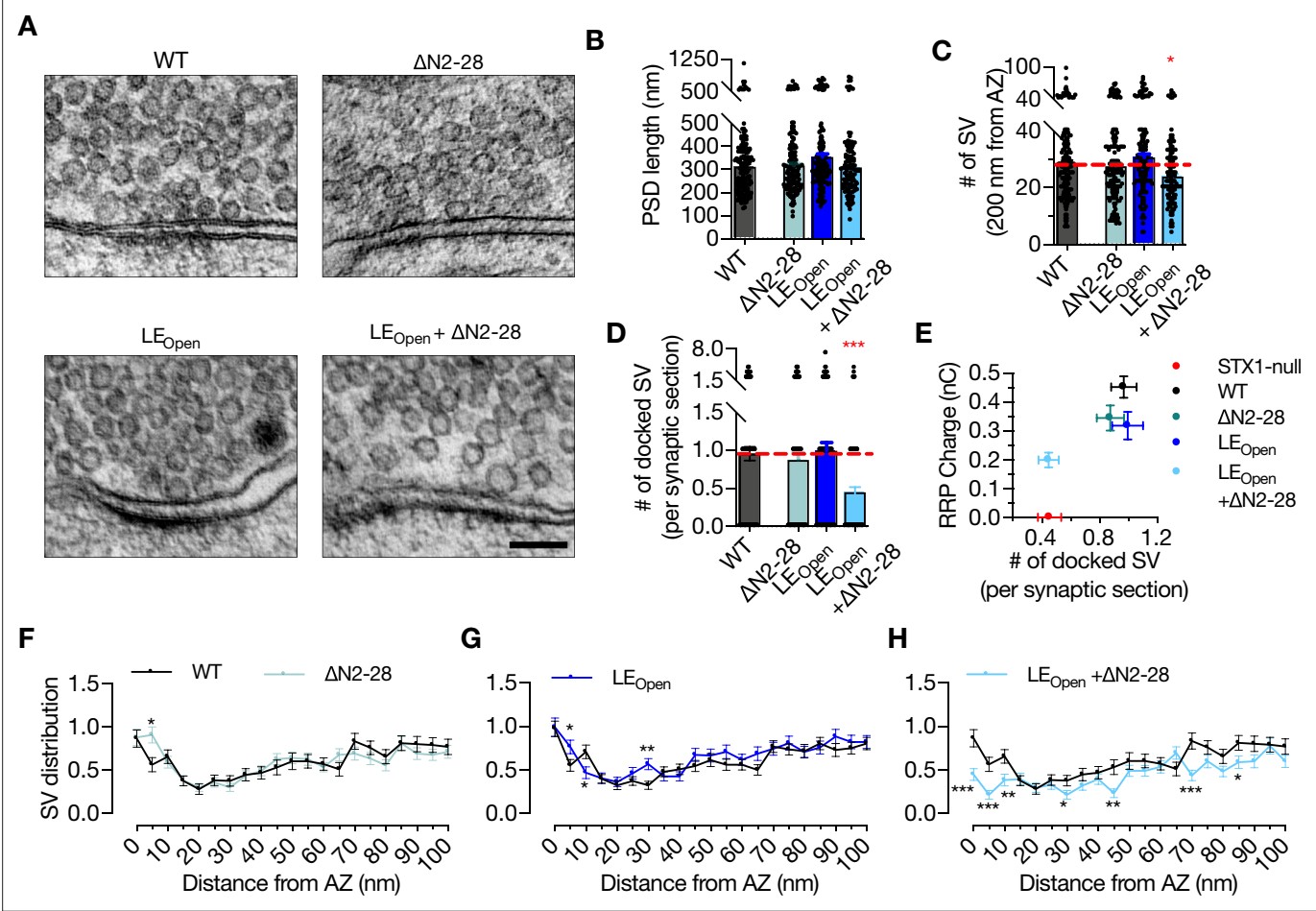

**Figure 5.** 'Opening' of STX1A in combination with the deletion of its entire N-terminal stretch reduces the number of docked synaptic vesicles (SVs). (**A**) Example high-pressure freezing fixation combined with electron microscopy (HPF-EM) images of nerve terminals from high-density cultures of STX1A[WT], STX1A[ΔN2-28], STX1A[LEOpen], and STX1A[LEOpen + ΔN2-28] Neurons. (**B–D**) Quantification of active zone (AZ) length, number of SVs within 200 nm distance from AZ, and number of docked SVs. (**E**) Correlation of the number of docked SVs obtained by HPF-EM to the size of readily releasable pool (RRP) obtained by electrophysiological recordings. (**F–H**) SV distribution of STX1A[ΔN2-28], STX1A[LEOpen], and STX1A[LEOpen + ΔN2-28] neurons compared to that of STX1A[WT] neurons. Data information: in (**B–D**), data points represent single observations, the bars represent the mean ± SEM. In (**E–H**), data points represent mean ± SEM. Black annotations on the graphs show the significance comparisons to STX1A[WT] rescue (nonparametric Kruskal–Wallis test followed by Dunn's *post hoc* test in **B–D**, multiple t-tests in **F–H** *p≤0.05, **p≤0.01, ***p≤0.001). The numerical values are summarized in *Figure 5—source data 1*.

The online version of this article includes the following source data and figure supplement(s) for figure 5:

**Source data 1.** Quantification of the ultrastructural synaptic properties of STX1A[WT], STX1A[ΔN2-28], STX1A[LEOpen], and STX1A[LEOpen + ΔN2-28] neurons.

**Figure supplement 1.** Synapse number and area are not affected neither by LE[Open] mutation nor by LE[Open] + ΔN mutation.

**Figure supplement 1—source data 1.** Quantification of the number the total area of VGlut1 positive puncta in autaptic neurons expressing STX1A[WT], STX1A[LEOpen], or STX1A[LEOpen + ΔN2-28].

It is known that Munc18-1 also functions upstream of the vesicle docking step (***Toonen et al., 2006***; ***Gulyas-Kovacs et al., 2007***). Therefore, we analyzed the state of docked vesicles in neurons that express either STX1A[ΔN2-28], STX1A[LEOpen], or STX1A[LEOpen+ΔN2-28] using HPF-EM (***Figure 5A***). PSD length, and thus AZ length, was again comparable between all the mutants and STX1A[WT] (***Figure 5B***), whereas STX1A[LEOpen+ΔN2-28] neurons showed a small but significant reduction in total SV number within 200 nm of AZ (***Figure 5C***). Strikingly, the neurons in which two Munc18-1 binding modes were modulated by the STX1A[LEOpen+ΔN2-28] mutation showed docked vesicles were reduced to ~50 % of those in STX1A[WT] synapses (***Figure 5D***). On the other hand, neither LE[Open] mutation nor N-peptide deletion alone did not influence vesicle docking (***Figure 5D***). Furthermore, vesicle distribution analysis revealed an accumulation of vesicles at 5 nm distance from AZ in STX1A[ΔN2-28] neurons but a reduction

in STX1A[LEOpen+ΔN2-28] neurons (*Figure 5F and G*), whereas STX1A[LEOpen] neurons did not show a major alteration in their vesicle distribution within 100 nm from AZ (*Figure 5H*).

It is possible that a reduction of RRP might reflect a reduction in synapse number if synapse loss precedes neuronal loss in the case of STX1A[LEOpen] and STX1A[LEOpen+ΔN2-28] neurons. In that scenario, number of docked vesicles, which is morphologically assessed by evaluating the existing synapses in mass culture, would not be affected and thus lead to differential outcomes for vesicle docking and vesicle fusion in neurons destined to death. To test this, we analyzed the synapse number in autaptic neurons for which we used the images shown in *Figure 4—figure supplement 2* and found no difference neither in the synapse number nor in the synapse area among STX1A[WT], STX1A[LEOpen], and STX1A[LEOpen+ΔN2-28] neurons as determined by VGlut1-positive puncta (*Figure 5—figure supplement 1*). Previously, we have shown that vesicle priming can be completely abolished by a STX1A mutant (A240V, V244A) with the vesicle docking remaining intact (*Vardar et al., 2016*). We also have reported that the vesicle priming is more prone to impairments by mutations in the vesicle release machinery than is vesicle docking, which suggests a separation or a different cooperativity between these events (*Zarebidaki et al., 2020*). In this light, we plotted the number of docked SVs versus the RRP size and observed that the RRP is also more susceptible to a reduction than is vesicle docking for the STX1A–Munc18-1 binding mutants (*Figure 5E*).

## STX1's N-peptide has a modulatory function in short-term plasticity and Ca²⁺-sensitivity of synaptic transmission

So far, our analysis has shown that STX1's N-peptide is not indispensable for neurotransmitter release (*Figures 1 and 3*), but plays a modulatory role in protein expression (*Figure 3*) and, when STX1's open conformation is facilitated by LE_Open mutation, in vesicle fusion and Pvr (*Figure 4*). To elucidate the modulation of neurotransmitter release by STX1's N-peptide, we took a closer look at Pvr and its effect on STP (*Figure 6A*). Even though the neurons expressing any STX1A[ΔN] mutants showed only a trend towards decreased Pvr compared to that of STX1A[WT] neurons (*Figure 3*), their STP behavior in response to 50 stimuli at 10 Hz differed greatly (*Figure 6A*). Both STX1A[ΔN2-19] and STX1A[ΔN2-28] showed first zero then only ~10 % depression following the first stimulus and STX1A[ΔN2-9] exhibited less depression than STX1A[WT] after the first 10 stimuli as analyzed by normalizing the EPSC responses to the first response (*Figure 6A*). Because STX1A[ΔN2-19] and STX1A[ΔN2-28] neurons have a reduced initial EPSC compared to that of STX1A[WT] neurons, we also plotted the absolute values of EPSCs elicited at 10 Hz (*Figure 6—figure supplement 1*). STX1A[ΔN2-28] tended to remain to elicit smaller EPSCs throughout the high-frequency stimuli (HFS) compared to those of STX1A[WT] (*Figure 6—figure supplement 1*). Zoomed-in example traces for the representation of the first and last five stimuli can be found in *Figure 6—figure supplement 2*.

Whereas Pvr shapes the STP curve starting from the initial phase, the late phase of STP is affected not only by Pvr, but also by the rate of SVs newly arriving at the AZ. Because all the STX1A[ΔN] neurons showed an altered behavior in the late phase of STP, we hypothesized that these neurons might keep up better with the high-frequency stimulus than STX1A[WT] does because of an increase in newly arrived SVs or replenishment of the RRP. To study the efficacy of replenishment of the primed vesicles, we stimulated the neurons with a double pulse of 500 mM sucrose solution with a time interval of 2 s (*Figure 6B*) as the replenishment of the whole pool of the primed vesicles after sucrose depletion takes at least 10 s (*Stevens and Tsujimoto, 1995*). STX1A[ΔN2-9] showed no effect on the fraction of the RRP recovered after full depletion when compared to that of STX1A[WT] (*Figure 6B*). On the other hand, STX1A[ΔN2-19] and STX1A[ΔN2-19] slightly decreased the vesicle replenishment rate by ~10%, which is contrary to our initial expectation (*Figure 6B*). This suggests that an increased replenishment rate does not account for the decreased depression in the STP curves of STX1A[ΔN] mutants.

Pvr and the degree of STP depend on both vesicle fusogenicity and Ca²⁺-sensitivity of the vesicular release, thus we scrutinized the effects of N-peptide deletions on these variables. Calculating the RRP fraction released in response to a sub-saturating, 250 mM sucrose solution application revealed no difference between STX1A[WT] and STX1A[ΔN] neurons, suggesting no decrease in SV fusogenicity with these mutants (*Figure 6C*). Interestingly, when we generated Ca²⁺-dose–response curves by evoking AP driven EPSCs in the presence of 0.5, 1, 2, 4, or 10 mM Ca²⁺-containing extracellular solutions, we observed a slightly lowered apparent Ca²⁺-sensitivity in the STX1A[ΔN2-19] (*Figure 6—figure supplement 1*) and STX1A[ΔN2-28] neurons (*Figure 6D*) when compared to STX1A[WT] neurons. On the other

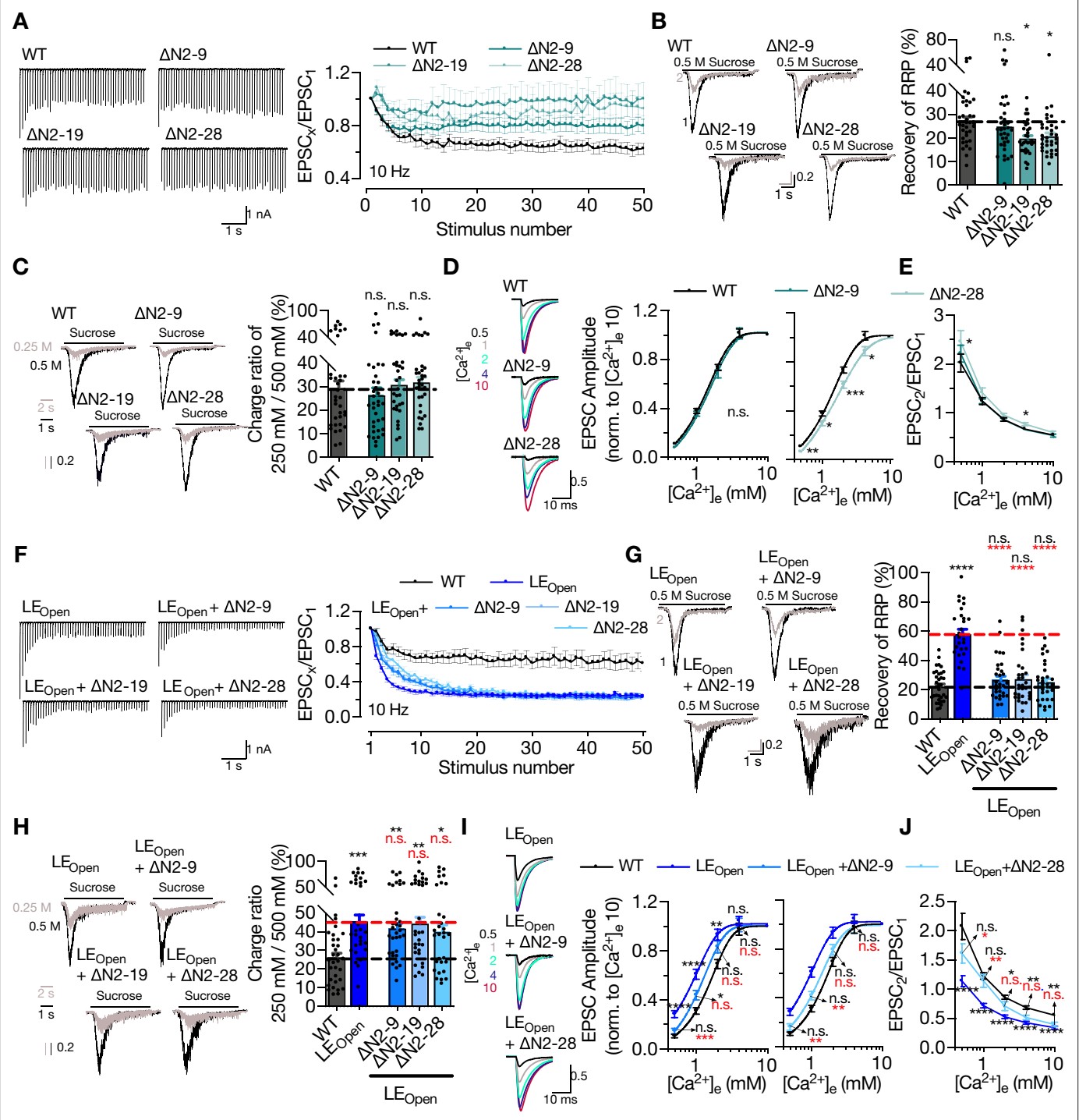

**Figure 6.** STX1A's N-peptide has a modulatory function in short-term plasticity and Ca²⁺-sensitivity of synaptic transmission. (**A**) Example traces (left) and quantification (right) of STP measured by 50 stimulations at 10 Hz from STX1A$^{WT}$, STX1A$^{\Delta N2-9}$, STX1A$^{\Delta N2-19}$, or STX1A$^{\Delta N2-28}$ neurons. The traces show the absolute values, whereas the quantification shows normalized EPSC to EPSC$_1$. (**B**) Example traces (left) and quantification (right) of the recovery of readily releasable pool (RRP) determined as the fraction of RRP measured at a second pulse of 500 mM sucrose solution after 2 s of initial depletion from STX1A$^{WT}$, STX1A$^{\Delta N2-9}$, STX1A$^{\Delta N2-19}$, or STX1A$^{\Delta N2-28}$ neurons. (**C**) Example traces (left) and quantification (right) of the ratio of the charge transfer triggered by 250 mM sucrose over that of 500 mM sucrose as a read-out of fusogenicity of the synaptic vesicles (SVs). (**D**) Example traces (left) and quantification (right) of Ca²⁺-sensitivity as measured by the ratio of EPSC amplitudes at [Ca²⁺]$_e$ of 0.5, 1, 2, 4, and 10 mM recorded from STX1A$^{WT}$, STX1A$^{\Delta N2-9}$, or STX1A$^{\Delta N2-28}$ neurons. The responses were normalized to the response at [Ca²⁺]$_e$ of 10 mM. (**E**) Paired-pulse ratio (PPR) of EPSC amplitudes at [Ca²⁺]$_e$ of 0.5, 1, 2, 4, and 10 mM recorded at 40 Hz. (**F**) Example traces (left) and quantification (right) of STP measured by 50 stimulations at 10 Hz from STX1A$^{WT}$,

*Figure 6 continued on next page*

*Figure 6 continued*

STX1A$^{LEOpen}$, STX1A$^{LEOpen + \Delta N2-9}$, STX1A$^{LEOpen + \Delta N2-19}$, or STX1A$^{LEOpen + \Delta N2-28}$ neurons. The traces show the absolute values, whereas the quantification shows normalized EPSC to EPSC$_1$. (**G**) Example traces (left) and quantification (right) of the recovery of RRP determined as the fraction of RRP measured at a second pulse of 500 mM sucrose solution after 2 s of initial depletion from STX1A$^{WT}$, STX1A$^{LEOpen}$, STX1A$^{LEOpen + \Delta N2-9}$, STX1A$^{LEOpen + \Delta N2-19}$, or STX1A$^{LEOpen + \Delta N2-28}$ neurons. (**H**) Example traces (left) and quantification (right) of the ratio of the charge transfer triggered by 250 mM sucrose over that of 500 mM sucrose as a read-out of fusogenicity of the SVs. (**I**) Example traces (left) and quantification (right) of Ca$^{2+}$-sensitivity recorded from STX1A$^{WT}$, STX1A$^{LEOpen}$, STX1A$^{LEOpen + \Delta N2-9}$, or STX1A$^{LEOpen + \Delta N2-28}$ neurons. The responses were normalized to the response at [Ca$^{2+}$]$_e$ of 10 mM. (**J**) PPR of EPSC amplitudes at [Ca$^{2+}$]$_e$ of 0.5, 1, 2, 4, and 10 mM recorded at 40 Hz from STX1A$^{WT}$, STX1A$^{LEOpen}$, or STX1A$^{LEOpen + \Delta N2-28}$ neurons. Data information: the artifacts are blanked in example traces in (**A, D, F, I**). In (**A, D, E, F, I, J**), data points represent the mean ± SEM. In (**B, C, G, H**), data points represent single observations, the bars represent the mean ± SEM. Black and red annotations on the graphs show the significance comparisons to STX1A$^{WT}$ or STX1A$^{LEOpen}$, respectively. (either nonparametric Kruskal–Wallis followed by Dunn's *post hoc* test or one-way ANOVA followed by Holm–Sidak's *post hoc* test was applied based on the normality of the data, *p≤0.05, **p≤0.01, ***p≤0.001, ****p≤0.0001). The numerical values are summarized in *Figure 6—source data 1*.

The online version of this article includes the following source data and figure supplement(s) for figure 6:

**Source data 1.** Quantification of the STP, recovery of RRP, RRP fraction released by 250 mM sucrose solution application and Ca2+-sensitivity of the vesicles in neurons expressing STX1A$^{WT}$, STX1A$^{\Delta N}$- or STX1A$^{LEOpen}$ mutants.

**Figure supplement 1.** Deletion of N-peptide increases the paired-pulse ratio (PPR) both in closed and open conformation of STX1A in low extracellular Ca$^{2+}$-concentration.

**Figure supplement 1—source data 1.** Quantification of the absolute values of EPSCs during STP and Ca$^{2+}$-sensitivity of the vesicles in neurons expressing STX1A$^{WT}$, STX1A$^{\Delta N}$- or STX1A$^{LEOpen}$ mutants.

**Figure supplement 2.** Zoomed-in example traces of STP.

hand, STX1A$^{\Delta N2-9}$ neurons showed a normal pattern of increase in EPSCs in relation to increasing extracellular Ca$^{2+}$-concentration (*Figure 6D*). We also measured the PPR, which is inversely related to Pvr, at different extracellular Ca$^{2+}$-concentrations and determined that STX1A$^{\Delta N2-28}$ had a significantly higher PPR at 0.5 mM [Ca$^{2+}$]$_e$ compared to that of STX1A$^{WT}$ (*Figure 6E*). This was also evident in STP behavior elicited by 5 AP stimulation at 40 Hz as STX1A$^{\Delta N2-28}$ neurons showed a greater facilitation at 0.5 mM [Ca$^{2+}$]$_e$ compared to that of STX1A$^{WT}$ (*Figure 6—figure supplement 1*). Whereas increasing [Ca$^{2+}$]$_e$ to 2 mM was not sufficient to drive the STP behavior of STX1A$^{\Delta N2-28}$ neurons towards STX1A$^{WT}$-like pattern, at the highest [Ca$^{2+}$]$_e$ tested all the groups STX1A$^{WT}$, STX1A$^{\Delta N2-9}$, and STX1A$^{\Delta N2-28}$ showed a similar level of depression upon 40 Hz stimulation (*Figure 6—figure supplement 1*).

It is well documented that the presumed facilitation of opening of STX1 and thus the increase in Pvr by LE$_{Open}$ mutation enhances short-term depression (*Acuna et al., 2014*; *Gerber et al., 2008*). Deletion of N-peptide at any length in STX1A$^{LEOpen}$ did not change the degree of the depression in the late phase of the high-frequency stimulus; however, all the deletions decreased the slope of the depression in the initial phase compared to that of STX1A$^{LEOpen}$ alone (*Figure 6F*). Whereas the EPSCs recorded from STX1A$^{LEOpen}$ neurons by 10 Hz stimulation tended to be initially larger, they declined further compared to that of STX1A$^{WT}$ neurons (*Figure 6—figure supplements 1 and 2*). On the other hand, STX1A$^{LEOpen+\Delta N2-28}$ mutants remained to produce smaller EPSCs throughout the HFS compared to those of both STX1A$^{WT}$ and STX1A$^{LEOpen}$ (*Figure 6—figure supplements 1 and 2*). The recovery of the RRP after sucrose depletion was enhanced by the open conformation of STX1A but reverted back to the WT-like levels by the expression of STX1A$^{LEOpen+\Delta N}$ mutants (*Figure 6G*). Strikingly, the increase in fusogenicity was not influenced by the N-peptide deletions (*Figure 6H*), which is consistent with the observation that those mutants also did not change LE$_{Open}$ mutation-driven enhancement of short-term depression at the late phase of the HFS (*Figure 6F*). On the other hand, N-peptide deletions imposed a right-shift in the Ca$^{2+}$-dose–response curves on the STX1A$^{LEOpen}$, which markedly increased the Ca$^{2+}$-sensitivity, making them approach again WT-like levels (*Figure 6I*, *Figure 6—figure supplement 1*). Consistent with increased fusogenicity and Ca$^{2+}$-sensitivity, STX1A$^{LEOpen}$ neurons always showed a greatly reduced PPR when compared to STX1A$^{WT}$ neurons in all extracellular Ca$^{2+}$-concentrations tested (*Figure 6J*). However, at low extracellular Ca$^{2+}$-concentrations STX1A$^{LEOpen+\Delta N2-28}$ neurons exhibited a PPR comparable to that of STX1A$^{WT}$ neurons, but at high Ca$^{2+}$-concentrations it was comparable to that of STX1A$^{LEOpen}$ neurons (*Figure 6J*). Contrary to STX1A$^{WT}$ neurons, STX1A$^{LEOpen}$ neurons showed no facilitation at 0.5 mM [Ca$^{2+}$]$_e$ and a greater depression at 2 mM [Ca$^{2+}$]$_e$ as well as at 10 mM [Ca$^{2+}$]$_e$. Whereas STX1A$^{LEOpen+\Delta N2-28}$ showed a similar pattern of facilitation at 0.5 mM [Ca$^{2+}$]$_e$ to STX1A$^{WT}$, at higher [Ca$^{2+}$]$_e$ their short-term depression approached the level of STX1A$^{LEOpen}$ (*Figure 6—figure supplement 1*). This and the observation that N-peptide deletion leads to an altered behavior only in

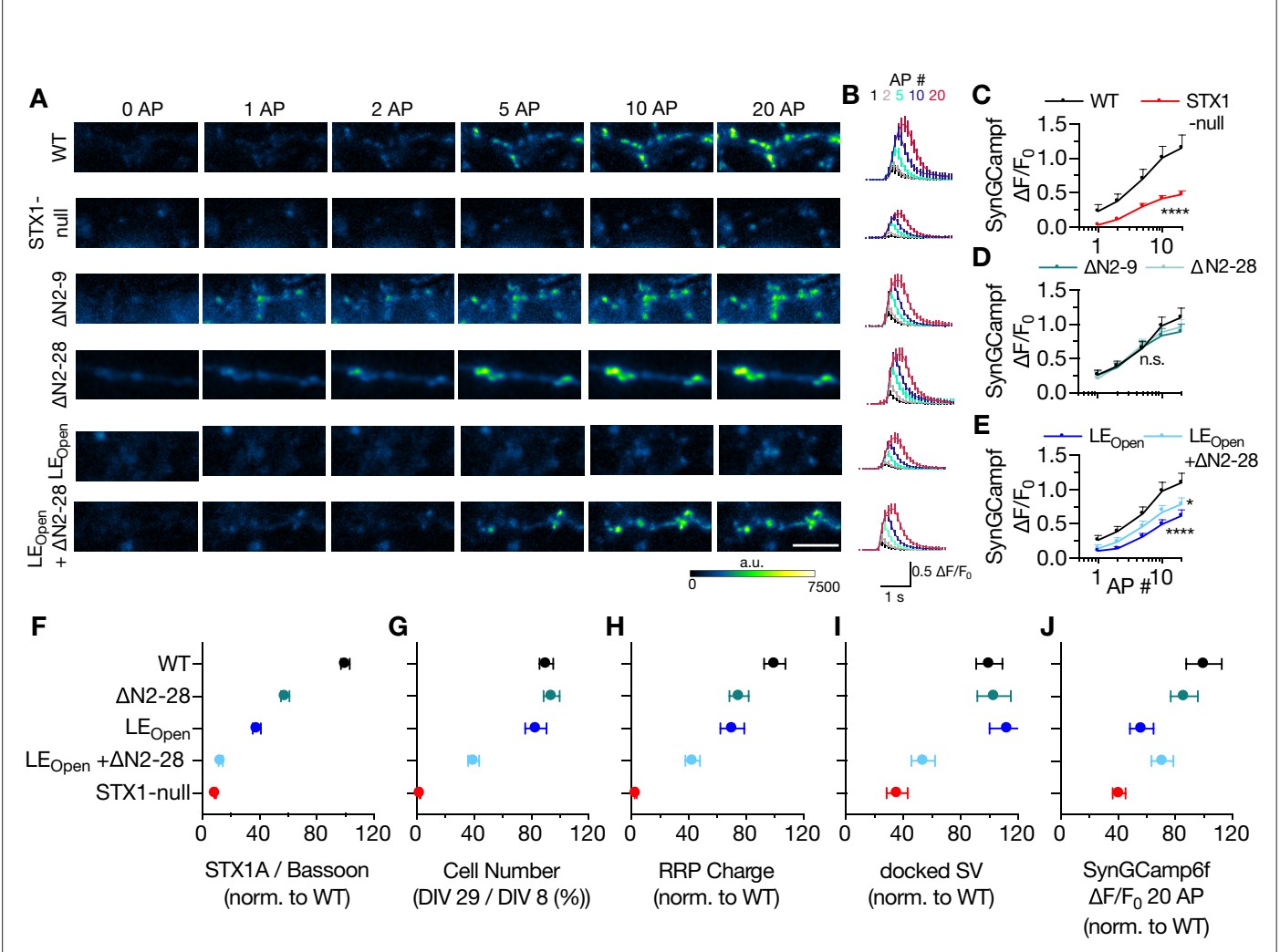

**Figure 7.** Ca²⁺-influx is reduced in STX1-null and in STX1$^{LEOpen}$ neurons. (**A, B**) Example images and average of SynGCaMP6f fluorescence as (ΔF/F₀) in STX1-null neurons either not rescued or rescued with STX1A$^{WT}$, STX1A$^{ΔN2-9}$, STX1A$^{ΔN2-19}$, STX1A$^{LEOpen}$, or STX1A$^{LEOpen + ΔN2-28}$. The images were recorded at baseline, and at 1, 2, 5, 10, and 20 action potentials (APs). Scale bar: 10 μm (**C–E**) Maximum fluorescence changes (ΔF/F₀) in STX1-null, STX1A$^{ΔN2-9}$, STX1A$^{ΔN2-28}$, STX1A$^{LEOpen}$, or STX1A$^{LEOpen + ΔN2-28}$ in comparison to that in STX1A$^{WT}$ neurons recorded at 1, 2, 5, 10, and 20 APs. (**F–I-J**) Summary plots of STX1A expression level, neuronal viability, readily releasable pool (RRP) charge, number of docked synaptic vesicles (SVs), and maximum SynGCaMP6f ΔF/F₀ at 20 AP from STX1-null, STX1A$^{WT}$, TX1A$^{ΔN2-28}$, STX1A$^{LEOpen}$, and STX1A$^{LEOpen + ΔN2-28}$. All the values were normalized to the one obtained from STX1A$^{WT}$ neurons in each individual culture. Data information: data points in all graphs represent the mean ± SEM. Black annotations on the graphs show the significance comparisons to STX1A$^{WT}$ (either unpaired t-test or Mann–Whitney test was applied in **C** based on the normality of the data; in **D** and **E**, nonparametric Kruskal–Wallis test followed by Dunn's *post hoc* test was applied, *p≤0.05, ****p≤0.0001). The numerical values are summarized in *Figure 7—source data 1*.

The online version of this article includes the following source data and figure supplement(s) for figure 7:

**Source data 1.** Quantification of the increase in SynGCaMP6f signal recorded at baseline or different numbers of APs in neurons expressing STX1A$^{WT}$, STX1A$^{ΔN}$- or STX1A$^{LEOpen}$ mutants.

**Figure supplement 1.** Reducing the expression level of STX1A$^{WT}$ does not alter Ca²⁺-influx.

**Figure supplement 1—source data 1.** Quantification of the increase in SynGCaMP6f signal recorded at baseline or different numbers of APs in neurons expressing STX1A$^{WT}$ at low level.

the initial phase of the 10 Hz stimuli – when STX1A's open conformation is facilitated – is consistent with the reduced Ca²⁺-sensitivity (*Figure 6I*) but unaltered fusogenicity (*Figure 6F*) of the vesicles.

Decreased Ca²⁺-sensitivity can arise from either reduced Ca²⁺-influx as a result of alterations in Ca²⁺-channel localization or gating, or from a disturbance in Ca²⁺-secretion coupling. To address this issue, we expressed the Ca²⁺-reporter GCamp6f coupled to Synaptophysin (SynGCamp6f) in

STX1-null neurons with or without STX1A rescue constructs and measured the immunofluorescence at the synapses at baseline or upon 1, 2, 5, 10, or 20 AP stimulation at 10 Hz (*Figure 7A and B*). Surprisingly, STX1-null neurons showed a decreased global $Ca^{2+}$-influx compared to neurons rescued with STX1A$^{WT}$ (*Figure 7C*). However, STX1A$^{\Delta N2-9}$ or STX1A$^{\Delta N2-28}$ did not influence the SynGCamp6f signal at any AP number elicited (*Figure 7D*), whereas global $Ca^{2+}$-influx was reduced in synapses in STX1A$^{LE-Open}$ and STX1A$^{LEOPen+\Delta N2-28}$ neurons (*Figure 7E*).

The reduction in $Ca^{2+}$-influx at the presynaptic terminals in STX1-null, STX1A$^{LEOpen}$ and STX1A$^{LE-OPen+\Delta N2-28}$ neurons compared to that of STX1A$^{WT}$ neurons is indicative of involvement of STX1 in the vesicular release processes upstream of vesicle docking (*Figure 7A–E*). As these STX1A mutants also showed severely decreased expression levels (*Figure 4*), we hypothesized that the synaptic structural properties might be affected by the expression level of STX1A. To test this, we measured the global $Ca^{2+}$-influx in neurons expressing low level of STX1A$^{WT}$ by using again 1/12th of the initial viral load and observed no effect of reduced expression level of STX1A on SynGCamp6f signal (*Figure 7—figure supplement 1*), which indicates a functional account of open conformation of STX1A for global $Ca^{2+}$-influx reduction rather than an expressional account (*Figure 7E*). As a summary, we plotted expression level of STX1A, neuronal viability, size of the RRP, number of docked SV, or the level of $Ca^{2+}$-influx at 20 AP in relation to N-peptide deletion and/or open conformation of STX1A (*Figure 7F–J*). Almost all the parameters showed a decreased degree of rescue by the expression of STX1A with combined mutation of LE$_{Open}$ and N-peptide deletion (*Figure 7F–J*), suggesting a cooperative function of STX1A's closed conformation and N-peptide.

## Discussion

The tight interaction between STX1 and Munc18-1 is not dictated through a single contact point but rather spans a large area both on STX1 and Munc18-1 (*Misura et al., 2000*), to which STX1's N-peptide and closed conformation largely contribute. Using our STX1-null mouse model system, we can draw several conclusions from mutant STX1 rescue experiments: (1) STX1's H$_{abc}$-domain is essential for the stability of STX1 and Munc18-1, and thus for neurotransmitter release and overall STX1 function; (2) STX1's N-peptide is dispensable for neurotransmitter release, but has a modulatory function for STX1's stability, for $Ca^{2+}$-sensitivity of vesicular release, and importantly for STP; and (3) neurotransmitter release can proceed even when the two interaction modes are presumably intervened by N-peptide deletions in conjunction with LE$_{Open}$ mutation in STX1A (*Figure 8*).

### STX1's H$_{abc}$-domain is essential for the overall function of STX1

The three helical H$_{abc}$-domain constitutes a major portion of STX1. As the main driving force for vesicle fusion is the zippering of the SNARE domains of STX1, SNAP25, and Syb2 (*Rizo and Sudhof, 2012*; *Rizo and Xu, 2015*; *Baker and Hughson, 2016*), STX1 that lacks only the H$_{abc}$-domain sufficiently mediates liposome fusion in reconstitution experiments (*Rathore et al., 2010*; *Shen et al., 2010*). Even in the synaptic environment, a crucial function of the H$_{abc}$-domain has been suggested only for spontaneous neurotransmitter release (*Zhou et al., 2013*; *Meijer et al., 2012*). However, the picture in intact synapses is more complex because neurotransmitter release proceeds as a result of multiple steps dependent on protein folding and trafficking, inter- and intramolecular interactions, relative conformations, and the proper localization of multiple synaptic proteins. Due to the lack of significant expression of STX1A$^{\Delta Habc}$ (*Figure 1*), we cannot draw a certain conclusion on whether or not H$_{abc}$-domain of STX1 is directly involved in neurotransmitter release. However, it has to be noted that even though Munc18-1 with a E259K point mutation, which interferes with its interaction with STX1's H$_{abc}$-domain as well as with the SNARE complex, could mediate $Ca^{2+}$-triggered neurotransmitter release in Munc18-1 knock-out neurons to some extent, more than half of the neurons were reported as synaptically silent (*Meijer et al., 2012*). Additionally, while liposome fusion can proceed without STX1's H$_{abc}$-domain (*Rathore et al., 2010*), the function of Munc18-1 as a template for SNARE complex formation (*Ma et al., 2013*; *Ma et al., 2015*) is greatly perturbed by the absence of STX1A's H$_{abc}$-domain (*Jiao et al., 2018*).

Overall, what is clear from our study and the previous studies is the importance of H$_{abc}$-domain in proper folding of STX1 and its co-recruitment to the AZ with Munc18-1. Severely decreased expression levels of STX1, Munc18-1, or both occur when the interaction between the H$_{abc}$-domain

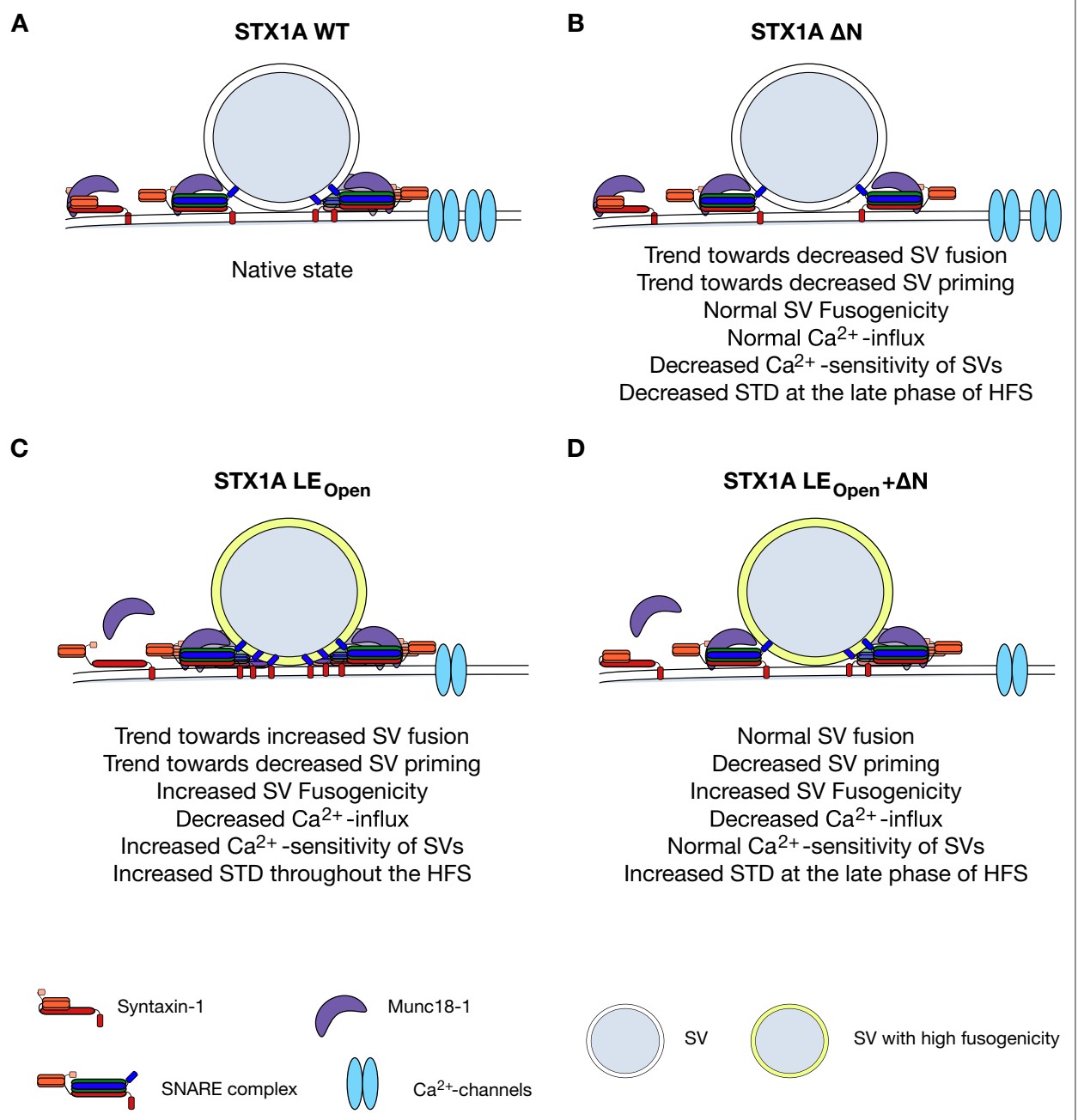

**Figure 8.** Speculative model of effects of N-peptide deletions and LE$_{Open}$ mutation on vesicular release. (**A**) Native state of STX1A. (**B**) N-peptide deletion of STX1A leads to a decrease in Ca$^{2+}$-sensitivity of vesicular release and short-term depression (STD) upon 10 Hz stimulation potentially through increased distance of Ca$^{2+}$-channel synaptic vesicle (SV) coupling. (**C**) LE$_{Open}$ mutation on STX1A increases fusogenicity and Ca$^{2+}$-sensitivity of SVs and thus leads to a high degree of STD. It also leads to reduced global Ca$^{2+}$-influx. (**D**) SV fusion proceeds normal when LE$_{Open}$ mutation is combined with N-peptide deletion. LE$_{Open}$ mutation dictates SV fusogenicity and Ca$^{2+}$-influx by increasing the former and decreasing the latter.

of STX1 and Munc18-1 is interrupted (*Gulyas-Kovacs et al., 2007*; *Meijer et al., 2012*; *Vardar et al., 2020*; *Zhou et al., 2013*). Even improper folding of the H$_{abc}$-domain by an insertion/deletion (InDel) mutation, identified in relation to epilepsy, leads to a high degree of STX1 instability (*Vardar et al., 2020*). Consistently, both the STX1B InDel mutant (*Vardar et al., 2020*) and STX1A$^{\Delta Habc}$ mutant (*Figure 2*) were incapable of sustaining neuronal viability. Thus, we argue that the major role of H$_{abc}$-domain of STX1 is to drive it into its correct folding and to recruit it together with Munc18-1 to the AZ.

# STX1's N-peptide role in neurotransmitter release is only detectable in STX1's LE$_{Open}$ configuration

On the contrary to the general view, we show here that the STX1's N-peptide is not indispensable for neurotransmitter release, but rather only modulates STX1's expression and the Ca$^{2+}$-sensitivity of SVs. Above all, the dispensability of STX1's N-peptide in vesicle fusion and particularly for proper recruitment of Munc18-1 to the AZ is consistent with the estimated contribution of N-peptide to the overall affinity of STX1 to Munc18-1, which is only minor (*Burkhardt et al., 2008*; *Christie et al., 2012*; *Colbert et al., 2013*).

Remarkably, the putative loss of two canonical interaction modes between STX1–Munc18-1 has little or no effect on synaptic transmission in general. This is not unprecedented as STX1A$^{\Delta N}$ and STX1A$^{LEOpen}$ mutants exhibit a largely unaltered binding affinity to Munc18-1 (*Burkhardt et al., 2008*). Additionally, these mutants have also been proposed to maintain the closed conformation when bound to Munc18-1 (*Colbert et al., 2013*; *Dawidowski and Cafiso, 2013*; *Lai et al., 2017*; *Wang et al., 2017*) potentially through additional contact points on STX1A including its SNARE motif (*Misura et al., 2000*; *Burkhardt et al., 2008*; *Liang et al., 2013*). Given that and the flexibility of STX1–Munc18-1 interaction, which induces large conformational changes on these proteins not only when STX1 is isolated but also when it enters the SNARE complex (*Jakhanwal et al., 2017*), it is conceivable that even for the STX1A$^{LEOpen+\Delta N}$ mutants a level of interaction between STX1–Munc18-1 must be retained.

Nevertheless, our analysis shows that even though both N-peptide deletion and LE$_{Open}$ mutation produce the same degree of reduction in the binding affinity of STX1 to Munc18-1 (*Burkhardt et al., 2008*; *Christie et al., 2012*; *Colbert et al., 2013*), it is the native conformation that commands the Munc18-1 recruitment and/or stability at the synapse (*Figures 1, 3 and 4*). This led us to interpret Munc18-1's binding to STX1 and its ultimate effect on SNARE complex formation as a two-step process, which is a convolution of the affinity and the efficacy of this interaction. Consistently, it has been thought that LE$_{Open}$ mutation exposes the SNARE domain of STX1 (*Dulubova et al., 1999*), whereas absence of N-peptide tightens its Munc18-1-driven closed conformation (*Khvotchev et al., 2007*; *Christie et al., 2012*; *Colbert et al., 2013*) potentially resulting in opposing effects in SNARE complex formation. Indeed, our observation that N-peptide deletion reverses the STX1A$^{LEOpen}$-dependent facilitation of neurotransmitter release parameters, which is generally attributed to its promotion of SNARE complex formation (*Dulubova et al., 1999*; *Gerber et al., 2008*, *Acuna et al., 2014*), hints at a reduction in the number of SNARE complexes formed by STX1A$^{LEOpen}$. However, N-peptide likely plays only a minor role in determining the equilibrium of open-closed conformations of STX1$^{WT}$ in a membranous environment when STX1's TMR is present (*Dawidowski and Cafiso, 2013*), and thus the modulation of neurotransmitter release by the N-peptide cannot be observed in STX1$^{WT}$ but can be only unmasked by the introduction of LE$_{Open}$ mutation.

Importantly, Munc18-1 does not bind only to STX1, but it is also thought to bind to the SNARE complex formed by STX1, SNAP-25, and Syb-2 (*Zilly et al., 2006*; *Dulubova et al., 2007*; *Shen et al., 2007*; *Burkhardt et al., 2008*) to provide a template as a scaffold together with Munc13 (*Ma et al., 2013*; *Ma et al., 2015*; *Lai et al., 2017*; *Wang et al., 2017*). Given that the SNARE complex formation is a dynamic process, which involves not only the assistance but also the protection by Munc18-1 and Munc13 against NSF-dependent dissociation (*Ma et al., 2013*; *He et al., 2017*), such a two-step process is also applicable for the back- and forward shift in the number of SNARE complexes. Furthermore, one of Munc13's primary functions is priming the vesicles (*Varoqueaux et al., 2002*) in addition to templating proper SNARE complex formation for which it assists opening of Munc18-1-bound STX1 (*Ma et al., 2011*; *Ma et al., 2013*; *Lai et al., 2017*; *Wang et al., 2017*). In this context, it has been shown that STX1A$^{LEOpen}$ recovers neurotransmitter release in Munc13-1/2-deficient mouse or worm neurons albeit only minimally (*Lai et al., 2017*; *Tien et al., 2020*), while it impairs Munc13's proper assistance of parallel SNARE complex formation (*Wang et al., 2017*) and further reduces the locomotor activity and neurotransmitter release in Munc18-1-deficient worms (*Tien et al., 2020*). Thus, it is plausible that the stability of the SNARE complex is ensured by Munc18-1's and Munc13's efficient binding to STX1, which may account for the additive effects of N-peptide deletion and LE$_{Open}$ conformation on the size of RRP (*Figures 4 and 7*). However, this regulation process must be upstream of the vesicle priming as the fusogenicity of the primed vesicles was predominantly dictated by the LE$_{Open}$ mutation.

# STX1A's N-peptide regulates Ca²⁺-sensitivity of SVs and short-term plasticity, whereas LE$_{Open}$ mutation dictates the SV fusogenicity

How the SVs become more fusogenic in the presence of STX1A$^{LEOpen}$ is not known, though one simple explanation is its propensity to produce reactive SNARE complexes with a higher number and efficacy (*Dulubova et al., 1999*; *Acuna et al., 2014*; *Gerber et al., 2008*). This hypothesis is appealing as it can also account for the faster recovery of the SVs to the RRP by LE$_{Open}$ mutation (*Figure 6*). Surprisingly, however, addition of N-peptide deletions not only in STX1A$^{LEOpen}$ but also in STX1A$^{WT}$ exclusively slowed down the RRP replenishment without an effect on the SV fusogenicity (*Figure 6*), uncoupling the regulation of these two processes. We cannot explain this phenomenon based on our data but would like to draw attention to that there are still unsolved questions regarding the regulation of SV fusogenicity. In fact, it is thought that at the state of primed and even docked vesicles the SNAREs are zippered up to hydrophobic layer +4 (*Sorensen et al., 2006*; *Walter et al., 2010*; *Vardar et al., 2016*) and thus include already 'opened' STX1. Therefore, the increase in SV fusogenicity by STX1A-$^{LEOpen}$- and STX1A$^{LEOpen+ΔN}$-mutants might involve yet an unknown mechanism, which does not employ STX1's N-peptide.

It is remarkable that the deletion of the N-peptide of 19 or 28 aa reduced the Ca²⁺-sensitivity of the vesicular release both in default and LE$_{Open}$ STX1A (*Figure 6*). Ca²⁺-sensitivity of the vesicular release is estimated by the assessment of Ca²⁺-dose–response, which is a convoluted measurement of SV fusogenicity and Ca²⁺-channel–SV distance coupling. Accordingly, the increased Ca²⁺-sensitivity of vesicular release by LE$_{Open}$ mutation stems partly from the increased fusogenicity of SVs (*Figure 6*). However, the rightward shift in Ca²⁺-dose–response curve caused by N-peptide deletions was not accompanied by an altered fusogenicity of SVs neither on default nor on LE$_{Open}$ background of STX1A. Thus, it is conceivable that the N-peptide deletions might have led to disrupted Ca²⁺-channel–SV coupling, without effecting the fusogenicity of the vesicles. Consistently, N-peptide deletions in the LE$_{Open}$ conformation, which do not alter LE$_{Open}$-dependent increase in vesicle fusogenicity, lead to an increase in PPR compared to that of LE$_{Open}$ mutation alone only at low external Ca²⁺-concentrations. This suggests that when the vesicles are positioned at a greater distance to Ca²⁺-channels in the absence of N-peptide, the enhancement of fusogenicity governed by the opening of STX1A by LE$_{Open}$ mutation remains insufficient to increase the Pvr at low Ca²⁺-concentrations. At higher Ca²⁺-concentration, on the other hand, the wider distance of the vesicles to Ca²⁺-channels becomes negligible and the LE$_{Open}$ mutation-dependent enhancement of vesicle fusogenicity dominates the Pvr and thus reduces PPR in the case of STX1A$^{LEOpen+ΔN}$ mutants (*Figure 6—figure supplement 1*) as speculatively illustrated in *Figure 8*. This can also explain why the N-peptide deletions on STX1A$^{LEOpen}$ background slow down depression during the initial phase of the 10 Hz stimuli, while they affect the STP only at the late phase on STX1A$^{WT}$ background (*Figure 6*).

Interestingly, our data reveal a new function of STX1 in synaptic transmission in that it controls global Ca²⁺-entry into the presynapse as STX1 deficiency and LE$_{Open}$ mutation led to a decreased Ca²⁺-influx (*Figure 7*). This can be explained either by alterations in Ca²⁺-channel gating and/or abundance. In fact, a direct interaction between STX1 and Ca²⁺-channels (*Bachnoff et al., 2013*; *Cohen et al., 2007*; *Sheng et al., 1994*; *Wiser et al., 1996*; *Sajman et al., 2017*) has been proposed to contribute to the overall Ca²⁺-sensitivity of the vesicular release machinery, where STX1 deemed an inhibitory role in baseline activity of Ca²⁺-channels (*Trus et al., 2001*). However, we observe a general decrease in Ca²⁺-entry also for higher numbers of AP elicited, not only after single AP (*Figure 7*). This is evocative of the phenotype of loss of RIMs, which are tethering factors of Ca²⁺-channels to the AZ (*Kaeser et al., 2011*; *Brockmann et al., 2020*). Thus, it is likely that decreased Ca²⁺-influx into the presynapse might be due to reduced number of Ca²⁺-channels at AZ in STX1-deficient neurons rather than due to an altered Ca²⁺-channel gating.

In fact, it is known that STX1 clusters together with Munc18-1 and SNAP25 also outside of AZ (*Pertsinidis et al., 2013*) and that it interacts with endoplasmic reticulum (ER) SNARE Sec22 at the ER-plasma membrane contact sites (*Petkovic et al., 2014*), both having potential functions in constitutive intracellular trafficking and regulation of the membrane lipid composition. In this regard, an impairment in general intracellular trafficking and/or membrane lipid composition as a result of loss of STX1, its presumed conformational change, and/or its deficient Munc18-1 binding imposed by LE$_{Open}$ mutation might potentially lead to a decreased number of Ca²⁺-channels. However, just as there is not always a direct correlation between the Ca²⁺-channel abundance and the level of Ca²⁺-entry, there is

not always a relationship between the Ca$^{2+}$-entry and the level of EPSC and Pvr. An example for that is the overexpression of Ca$^{2+}$-channel subunit α2δ also leading to overexpression of Ca$^{2+}$-channels in the synapse (*Hoppa et al., 2012*). In these synapses, surprisingly however, a reduction in Ca$^{2+}$-influx has been observed together with an increase both in EPSC and Pvr – similar to the phenotype of STX1A$^{LEOpen}$ neurons (*Hoppa et al., 2012*). Therefore, it is plausible that the increased SV fusogenicity might overcome the effect of low Ca$^{2+}$-entry into the synapse and still lead to an increased EPSC and Pvr when the SVs are localized at a proper distance to the Ca$^{2+}$-channels in STX1A$^{LEOpen}$ neurons (*Figure 8*). Plausibly, on the other hand, if increased SV fusogenicity is accompanied by a greater SV-Ca$^{2+}$-channel distance as thought for the case of STX1A$^{LEOpen + ΔN}$ neurons, the Ca$^{2+}$-sensitivity and the amplitude of EPSCs might approach back to the WT-like levels (*Figure 8*).

Together, our data suggest that even though deletion of N-peptide potentially reduces the number of reactive SNARE complexes (*Burkhardt et al., 2008*), which could explain the slower rate of the recovery of the RRP in neurons that express STX1A$^{ΔN2-19}$ or STX1A$^{ΔN2-28}$ (*Figure 6*), the level of this reduction appears to be not enough to decrease the baseline neurotransmitter release in STX1A$^{WT}$ (*Figure 3*) but only in STX1A$^{LEOpen}$ (*Figure 4*). The increase in the apparent Ca$^{2+}$-sensitivity of the vesicular release in STX1A$^{LEOpen}$ neurons has also been attributed to the increased number of SNARE complexes (*Acuna et al., 2014*), yet the increased fusogenicity of the SVs in those neurons beclouds this hypothesis. Whether or not reduced number of SNARE complexes can lead to the robust effect of N-peptide deletions on STP is not clear, but likely, as the longer N-peptide deletions showed a trend towards smaller absolute values of EPSCs throughout the high-frequency stimulus (*Figure 6—figure supplement 1*). Since N-peptide mutations do not significantly change the initial PPR, the effect during 10 Hz trains can also be explained by an impaired Ca$^{2+}$-channel–vesicle distance coupling and an accumulation of global Ca$^{2+}$ during the train. Therefore, whether the facilitation–hindrance of SNARE complex formation leads to changes in STP and/or Ca$^{2+}$-sensitivity of vesicular release should be investigated in depth. However, STX1A$^{ΔN2-9}$ neurons did not show any difference in the initial EPSCs but only in the latter phase of STP by up to ~30 % larger EPSCs compared to that of STX1A$^{WT}$ neurons (*Figure 6—figure supplement 1*) and also no difference in the Ca$^{2+}$-sensitivity (*Figure 6*) of the vesicular release. This suggests that the regulation of STP might be an important function of STX1A's N-peptide aa 2–9 independent of Munc18-1 because Munc18-1 mutants that cannot bind to the STX1's N-peptide do not manifest any regulatory effect on STP (*Meijer et al., 2012*).

## Materials and methods

### Animal maintenance and generation of mouse lines

All procedures for animal maintenance and experiments were in accordance with the regulations of and approved by the animal welfare committee of Charité-Universitätsmedizin and the Berlin state government Agency for Health and Social Services under license number T0220/09. The generation of STX1-null mouse line was described previously (*Arancillo et al., 2013*; *Vardar et al., 2016*).

### Neuronal cultures and lentiviral constructs

Hippocampal neurons were obtained from mice of either sex at postnatal day (P) 0–2 and seeded on the already prepared continental or micro-island astrocyte cultures as described previously (*Xue et al., 2007*; *Vardar et al., 2016*). The neuronal cultures were then incubated for 13–20 DIV in NeurobasalA supplemented with B-27 (Invitrogen), 50 IU/ml penicillin and 50 µg/ml streptomycin at 37 °C before experimental procedures. Neuronal cultures for EM and Ca$^{2+}$-influx and those for neuronal viability, immunofluorescence labeling, and electrophysiology experiments were transduced with lentiviral particles at DIV 2–3 and DIV 1, respectively. Lentiviral particles were provided by the Viral Core Facility (VCF) of the Charité-Universitätsmedizin, Berlin, and were prepared as previously described (*Vardar et al., 2016*). The cDNA of mouse STX1A (NM_016801) was cloned in frame after an NLS-GFP-P2A sequence within the FUGW shuttle vector (*Lois et al., 2002*) in which the ubiquitin promoter was replaced by the human synapsin 1 promoter (f(syn)w). The improved *Cre* recombinase (iCre) cDNA was c-terminally fused to NLS-RFP-P2A. SynGCamp6f was generated analogous to synGCamp2 (*Herman et al., 2014*), by fusing GCamp6f (*Chen et al., 2013*) to the C terminus of synaptophysin and within the f(syn)w shuttle vector (*Grauel et al., 2016*).

## Neuronal viability

The in vitro viability of the neurons was defined as the percentage of the number of neurons alive at DIV15, 22, 29, 36, and 43 compared to the number of neurons at DIV 8. Phase-contrast bright-field images and fluorescent images with excitation wavelengths of 488 and 555 nm were acquired with a DMI 400 Leica microscope, DFC 345 FX camera, HCX PL FLUOTAR 10 objectives, and LASAF software (all from Leica). Fifteen randomly selected fields of 1.23 $mm^2$ per well and two wells per group in each culture were imaged at different time points and the neurons were counted offline with the 3D Objects Counter function in Fiji software as described previously (*Vardar et al., 2016*). Sample size estimation was done as previously published (*Vardar et al., 2016*). MAP2 immunofluorescence labeling as shown in the figures is used only for representative purposes.

## Immunocytochemistry

The high-density cultured *STX1A*$^{-/-}$; *STX1B*$^{flox/flox}$ hippocampal neurons were co-transduced with *Cre* recombinase and with either STX1A$^{WT}$ or mutants at DIV 1–2. All the cultures were fixed with 4 % paraformaldehyde (PFA) in 0.1 M phosphate-buffered saline, PH 7.4, for 10 min at DIV14-16. The neurons were then permeabilized with 0.1 % Tween–20 in PBS (PBST) for 45 min at room temperature (RT) and then blocked with 5 % normal goat serum (NGS) in PBST. Primary antibodies were applied overnight at 4 °C and subsequently secondary antibodies were applied for 1 hr at RT in the dark. High-density hippocampal cultures of $50 \times 10^3$ seeded neurons for neuronal viability analysis were treated with chicken polyclonal anti-MAP2 (1:2000; M2694; Merck Millipore) and then with Alexa Fluor (A) 647 donkey anti-chicken IgG (Jackson ImmunoResearch). High-density hippocampal cultures of $25 \times 10^3$ seeded neurons for protein expression analysis were treated with guinea pig polyclonal anti-Bassoon (1:1000; Synaptic Systems), mouse monoclonal anti-STX1A (1:1000; Synaptic Systems), and rabbit polyclonal Munc18-1 (1:1000; Sigma-Aldrich) and then with rhodamine red donkey anti-guinea pig IgG, A488 donkey anti-mouse IgG, and A647 donkey anti-rabbit IgG (all from Jackson ImmunoResearch). Autaptic neurons were treated with guinea pig polyclonal anti-VGlut1 (1:4000; Synaptic Systems), mouse monoclonal anti-STX1A (1:1000; Synaptic Systems), and rabbit polyclonal Munc18-1 (1:1000; Sigma-Aldrich) and subsequently with rhodamine red donkey anti-guinea pig IgG, A647 donkey anti-mouse IgG (both from Jackson ImmunoResearch), and Pacific-Blue goat anti-rabbit IgG (ThermoFisher). All secondary antibodies were diluted in 1:500 in PBST. The coverslips were mounted on glass slides with Mowiol mounting agent (Sigma-Aldrich). The images were acquired with an Olympus IX81 epifluorescence-microscope with MicroMax 1300YHS camera using MetaMorph software (Molecular Devices). Exposure times of excitations were kept constant for each wavelength throughout the images obtained from individual cultures. Data were analyzed offline with ImageJ as previously described (*Vardar et al., 2016*). Sample size estimation was done as previously published (*Vardar et al., 2016*). The number of synapses was analyzed by using Object Analyzer macro plug-in in ImageJ.

## Electrophysiology

The hippocampal autaptic neurons were co-transduced with *Cre* recombinase and with either STX1A$^{WT}$ or mutants at DIV 1–3. Sample size estimation was done as previously published (*Rosenmund and Stevens, 1996*). Whole-cell patch-clamp recordings were performed on glutamatergic autaptic hippocampal neurons at DIV 14–20 at RT with a Multiclamp 700B amplifier and an Axon Digidata 1550B digitizer controlled by Clampex 10.0 software (both from Molecular Devices). The recordings were analyzed offline using Axograph X Version 1.7.5 (Axograph Scientific).

Prior to recordings, the transduction of the neurons was verified by RFP and GFP fluorescence. Membrane capacitance and series resistance were compensated by 70 %, and only the recordings with a series resistance smaller than 10 MΩ were used for further recordings. Data were sampled at 10 kHz and filtered by low-pass Bessel filter at 3 kHz. The standard extracellular solution was applied with a fast perfusion system (1–2 ml/min) and contained the following: 140 mM NaCl, 2.4 mM KCl, 10 mM HEPES, 10 mM glucose, 2 mM $CaCl_2$, and 4 mM $MgCl_2$ (300 mOsm; pH 7.4). Borosilicate glass patch pipettes were pulled with a multistep puller, yielding a final tip resistance of 2–5 MΩ when filled with KCl-based intracellular solution containing the following: 136 mM KCl, 17.8 mM HEPES, 1 mM EGTA, 4.6 mM $MgCl_2$, 4 mM ATP-$Na_2$, 0.3 mM GTP-$Na_2$, 12 mM creatine phosphate, and 50 U/ml phosphocreatine kinase (300 mOsm; pH 7.4).

The neurons were clamped at –70 mV in steady state. To evoke EPSCs, the neurons were depolarized to 0 mV for 2 ms. The size of the RRP was determined by a 5 s application of 500 mM sucrose in standard external solution (*Rosenmund and Stevens, 1996*) and the total charge transfer was calculated as the integral of the transient current. Fusogenicity measurement was conducted by application of 250 mM sucrose solution for 10 s and calculation of the ratio of the charge transfer of the transient current over RRP. For the analysis of the RRP recovery, a paired pulse of 5 s long 500 mM sucrose was applied with a time interval of 2 s. Spontaneous release was determined by monitoring mEPSCs for 30–60 s at –70 mV. To correct false-positive events, mEPSCs were recorded in the presence of 3 µM AMPA receptor antagonist NBQX (Tocris Bioscience) in standard external solution. The spontaneous release rate was assessed by the division of the mEPSC frequency over the number of primed vesicles to determine the fraction of the RRP released per second by spontaneous release.

For $Ca^{2+}$-sensitivity assays, EPSCs were evoked in extracellular solution containing 1 mM $MgCl_2$ and either 0.5, 1, 2, 4, or 10 mM $CaCl_2$. In between the test extracellular solution applications, standard extracellular solution was applied. For each concentration, six APs were elicited at 0.2 Hz. To control for rundown and cell-to-cell variability, the test solutions were applied either in increasing or decreasing concentration order for equal number of neurons and test responses were normalized to average EPSCs priorly recorded in standard external solution. Normalized responses were then normalized to the response in 10 mM $CaCl_2$. The normalized values were fitted into a standard Hill equation.

## SynGcamp6f-imaging

Imaging experiments were performed at DIV 13–16 on autapses in response to a single stimulus and stimuli trains of 10 Hz as described previously for SynGcamp2-imaging (*Herman et al., 2014*). Images were acquired using a 490 nm LED system (pE2; CoolLED) at a 5 Hz sampling rate with 25 ms of exposure time. The acquired images were analyzed offline using ImageJ (National Institute of Health), Axograph X (Axograph), and Prism 8 (GraphPad, San Diego, CA). Sample size estimation was done as previously published (*Herman et al., 2014*).

## High-pressure freezing and electron microscopy

The high-density cultured hippocampal neurons were co-transduced with *Cre* recombinase and with either STX1 WT or mutants at DIV 2–3 and high-pressure fixed under a pressure of 2100 bar in standard extracellular recording solution using an HPM 100 Leica or ICE Leica high-pressure freezer at DIV 14–16. Samples were then transferred into cryovials containing 1 % glutaraldehyde, 1 % osmium tetroxide, 1 % $ddH_20$ (Millipore) in anhydrous acetone and processed in an AFS2 automated freeze-substitution device (Leica) followed by a temperature ramp from –90 °C for 5 hr, from –90 °C to –50 °C (8 °C/hr), from –50 °C to –20 °C (6 °C/hr), for 12 hr at –20 °C, from –20 to +20 °C (10 °C/hr). After the freeze-substitution step, samples were embedded in epoxy epon 812 (EMS). Finally, samples were placed into capsules filled with pure epoxy epon 812 and further polymerized for 48 hr at 60 °C. Randomly selected areas of ~250 µm$^2$ containing neurons were ultracut into 40 nm thick slices using an Ultracut UCT ultramicrotome (Leica) and collected on 0.5 % formvar-coated 200-mesh copper grids (EMS). Those sections were contrasted with 0.1 % uranyl acetate for 1 hr and lead citrate (0.15 lead citrate, 0.12 m sodium citrate in $ddH_2O$). Images were collected blindly in an FEI Tecnai G20 electron microscope operating at 200 keV and digital images taken with a Veleta 2k × 2k CCD camera (Olympus). Synapses were analyzed blindly using an analysis program developed for ImageJ and MATLAB (*Watanabe et al., 2013*). AZs were defined as the membrane stretch directly opposite to the postsynaptic density, and docked vesicles were defined as those in direct contact with the plasma membrane. SV distribution was analyzed by calculating the shortest distance of each vesicle to the AZ membrane and binned with 5 nm. Sample size estimation was done as published previously (*Watanabe et al., 2013*).

## Statistical analysis

Data in bar graphs present single observations (points) and means ± standard error of the mean (SEM; bars). Data in x–y plots present means ± SEM. All data were tested for normality with Kolmogorov–Smirnov test. Data from two groups with normal or nonparametric distribution were subjected to Student's two-tailed t-test or Mann–Whitney nonparametric test, respectively. Data from more

than two groups were subjected to Kruskal–Wallis followed by Dunn's *post hoc* test when at least one group showed a nonparametric distribution. For data in which all the groups showed a parametric distribution, one-way ANOVA test followed by Tukey's *post hoc* test was applied. For STP measurements, two-way ANOVA test was used. All the tests were run with GraphPad Prism 8.3, and all the statistical data are summarized in the corresponding source data tables.

## Acknowledgements

We thank the Charité Viral Core facility, Katja Pötschke, and Bettina Brokowski for virus production, Berit Söhl-Kielczynski and Heike Lerch for technical assistance, Melissa Herman and Marcial Camacho for their contribution to the manuscript, and to all the Rosenmund Lab members for the discussions. This work was supported by the German Research Council (DFG) grants 388271549, 399894546, 436260754, and 278001972.

## Additional information

### Funding

| Funder | Grant reference number | Author |
| --- | --- | --- |
| Deutsche Forschungsgemeinschaft | SFB958 TRR186 | Christian Rosenmund |
| Deutsche Forschungsgemeinschaft | Reinhart Koselleck Projects | Christian Rosenmund |

The funders had no role in study design, data collection and interpretation, or the decision to submit the work for publication.

### Author contributions

Gülçin Vardar, Conceptualization, Data curation, Formal analysis, Investigation, Validation, Visualization, Writing – original draft, Writing – review and editing; Andrea Salazar-Lázaro, Marion Weber-Boyvat, Sina Zobel, Victor Wumbor-Apin Kumbol, Formal analysis, Investigation; Marisa Brockmann, Investigation; Thorsten Trimbuch, Resources; Christian Rosenmund, Conceptualization, Funding acquisition, Methodology, Project administration, Supervision, Writing – review and editing

### Author ORCIDs

Gülçin Vardar http://orcid.org/0000-0002-5295-1591
Marisa Brockmann http://orcid.org/0000-0002-1386-5359
Christian Rosenmund http://orcid.org/0000-0002-3905-2444

### Ethics

All procedures for animal maintenance and experiments were in accordance with the regulations of and approved by the animal welfare committee of Charité-Universitätsmedizin and the Berlin state government Agency for Health and Social Services under license number T0220/09. The generation of STX1-null mouse line was described previously (Arancillo et al. 2013, Vardar et al. 2016).

### Decision letter and Author response

Decision letter https://doi.org/10.7554/eLife.69498.sa1
Author response https://doi.org/10.7554/eLife.69498.sa2

## Additional files

### Supplementary files

• Transparent reporting form

### Data availability

All data generated or analysed during this study are included in the manuscript and supporting files. We uploaded source data files which show summary tables of mean, SEM, median, number of

independent cultures, number of independent measurements, real p value for each test performed, and statistical test used for each separate figure.

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
