## [Decision Letter]

**Acceptance summary:**

In this study Vardar et al., use patch-clamp electrophysiology in autaptic neurons to provide a systematic analysis of the different roles of the N-terminal domains and different Munc18 binding modes of Syntaxin-1. The complexity of the interaction between Munc18 and syntaxin and their isoforms in other species has led to a vexing complexity involving different interactions that appear to range from essential to dispensable in different experiments and contexts. This study compares many aspects of these interactions in the same neuronal system. The paper confirms previous observations, but also arrives at new conclusions. The authors show that the Habc-domain is essential for syntaxin's role in neurotransmitter release, while the N-peptide has a modulatory role. Disrupting both binding modes of Syntaxin-1 with M18 leads to strongly reduced levels of both proteins while neurotransmitter release can still occur.

**Decision letter after peer review:**

Thank you for submitting your article "Reexamination of N-terminal domains of Syntaxin-1 in vesicle fusion from central murine synapses" for consideration by *eLife*. Your article has been reviewed by 3 peer reviewers, and the evaluation has been overseen by a Reviewing Editor and Gary Westbrook as the Senior Editor. The reviewers have opted to remain anonymous. The reviewers have discussed their reviews with one another, and the Reviewing Editor has drafted this to help you prepare a revised submission.

Essential Revisions (for the authors):

1. The authors attempt to address the reduced protein expression by titrating down WT levels by using only 1/12 of the viral load. Strikingly, reducing WT STX1A levels by almost 6-fold had no effect on synaptic transmission. This raises important questions. At which point does STX1A become rate-limiting? And how do these protein levels compare to endogenous STXA levels? The authors should show the expression levels of their mutants relative to endogenous STX levels. Furthermore, a previous paper from the same group concluded that titrating Synatxin1 did impact synaptic transmission (Arancillo et al., J Neurosci 2013). The authors should discuss how their expression levels and findings compare to the findings in their previous paper.

2. While the effect of the N-peptide on most release parameters is absent/small, a robust effect was observed on short-term plasticity: up to70% larger EPSC's at the end of a 10Hz train in N-peptide mutants. This substantial effect is undervalued by the authors. Regulation of STP could be an important function of this domain. The authors mention this effect only briefly in the abstract, but should discuss this biological relevance in the Discussion. Second, typical traces of trains in Figure 6 should be of higher resolution and zoomed in more to critically evaluate. For instance, the standing (asynchronous) current that usually develops during 10Hz trains in autapses seems to be absent. Why is this the case? Since N-peptide mutations do not change PPR, the most logical explanation of the effect during 10Hz trains is a difference in the accumulation of global Ca during the train. This is plausible given the (disputed) interaction of syntaxin with calcium channels. Ideally, the experiments should be repeated +/- EGTA-AM. In this same experiment, the author can examine the potential effect on Ca-dependent replenishment or increase in fusogenicity (measuring RRP size and fusogenicity before and after 10Hz trains). At the minimum, the observed effect should be more pointed out more clearly in the text.

3. The main hypothesis put forward for reduced Ca-sensitivity and impaired RRP recovery in N-peptide mutants is an impairment in SNARE complex formation. How does this reconcile with up to 70% larger EPSC's during 10 Hz trains? Please comment and discuss.

4. The data in figure 6 on the interaction between the two M18 binding modes of Stx1 are intriguing. However, different effects of N-peptide mutations in the WT and LE_Open background are difficult to reconcile, and not further addressed in the discussion. It is therefore difficult to interpret these data in the light of the N-peptide function. Please comment and discuss.

5. In Figure 6F the N-peptide mutations in Stx1 LE_Open slow down depression during the first few stimulations, but have no effect on STP at the end of the train. This is opposite from the effect found at the WT background, where these mutations affect STP at the end of the train (Figure 6A). How these seemingly different results can be reconciled should be addressed in the discussion.

6. Given the twofold higher EPSC amplitude and Pvr for Stx1 LE_Open, this mutant will most likely end up with similar EPSC amplitudes at the end of the train as for the WT. However, the LE_Open Δ_N mutants will have lower EPSC amplitudes at the end of the train. This could lead to different conclusions about the role of the N-peptide, especially when combined with similar experiments with these mutants as suggested in point 2.

7. It is interesting to see in Figure 6J that, for low external Ca, N-peptide mutations on a LE_open background rescue the paired-pulse ratio to WT levels, while for increasing external Ca levels this gradually reverts to the LE_open phenotype. This should be addressed in the discussion in the light of the different Ca sensitivities and fusogenicities for the different experimental groups.

8. If reduced Ca-influx explains reduced Ca affinity in LE_open mutant (Figure 6I), how does this reconcile with the LE_open-deltaN mutant, which has significant less Ca influx compared to WT (Figure 7E), but no significant difference in Ca sensitivity? Some explanation should be added to the Discussion.

9. The authors' interpretation that the effect of calcium influx relates to expression levels and survivability is somewhat challenged by the fact that the LE mutant has the strongest effect on calcium influx, while it is by far not the mutant with the lowest expression levels, nor is it experiencing a decrease in survivability. Please comment and discuss.

10. Throughout the manuscript, the authors quantify cell numbers as a measure of survival. Some mutations, for example syntaxin knockout and the DeltaHabc mutant rescue, strongly affect cell survival. For the interpretation of the synaptic physiology, a more meaningful parameter would be synapse numbers, especially for evaluating the relationship between docking (measured per synapse profile) and sucrose RRP (measured per cell). It appears possible, if not likely, that synapses are lost before cells die, and in this case correlating RRP and docking may be strongly confounded. The authors may have the data in mass cultures already, and it should be possible to quantify synapse numbers (for example via Bassoon puncta densities) in these existing data. This should be accompanied by a discussion of the systems used (mass cultures for morphology and autaptic cultures for electrophysiology). If synapse numbers are changed, the RRP charge and # of docked vesicles should not be directly correlated (as done in Figure 5E), because RRP charge strongly depends on synapse numbers, while the number of docked vesicles per synapse does not. The discussion should be adjusted accordingly.

11. The authors present the double mutation of opening syntaxin (LE mutant) and N-peptide deletion as "interruption of both Munc18-1 binding modes" (caption line 185, figure captions Figures 4 and 5). While we appreciate where the terminology for considering this a "null for Munc18 binding" comes from, it would be much better to stay closer to the experiments in the data presentation and use wording like "opening of syntaxin combined with N-peptide deletion" instead. As the authors discuss, Munc18-syntaxin interactions are complex, and most data suggest that Munc18 also binds to SNARE complexes (and open syntaxin) while they are being assembled. Opening syntaxin and deleting the N-peptide does unlikely fully abolish Munc18 binding, but biases binding of syntaxin to Munc18 alone towards binding of Munc18 to syntaxin as it is being incorporated into SNARE complexes. The authors should do justice to this point in figure captions, and should discuss it as such.

12. The effect of the LE mutant of syntaxin has been characterized by several biophysical methods that show that it shifts the conformational equilibrium towards the open state, but it still adopts the closed state upon Munc18 binding as observed by SAXS (Colbert et al. 2013) and single molecule FRET (Wang et al., EMBO J. 36, 816-, 2017). Moreover, the LE mutant only partially compensates for deletion of Munc13, i.e., the rescue is rather limited as observed in autaptic neuronal cultures and by single molecule FRET experiments (Lai et al., Neuron 95, 591-, 2017). Moreover, the results in Wang et al., 2017 suggest that the LE mutant interferes with proper Munc13 function (via its interactions with syntaxin), which may be related to the observed phenotypes of this mutant. Please discuss the experiments presented here in the context of these biophysical findings.*Reviewer #1 (Recommendations for the authors):*

In this study Vardar et al., use patch-clamp electrophysiology in autaptic neurons to provide a systematic analysis of the different roles of the N-terminal domains and different Munc18 binding modes of Syntaxin-1. The complexity of the interaction between Munc18 and Syntaxin has led to a long-standing controversy about the importance of certain interactions, ranging from absolutely essential to dispensable. This study is the first to compare all aspects in a side-by-side comparison in the same, native model system. The paper confirms previous observations but also arrives at new conclusions. The authors show that the Habc-domain is essential for Stx1's role in neurotransmitter release, while the N-peptide has a minor modulatory role. Disrupting both binding modes of Stx1 with M18 leads to strongly reduced levels of both proteins while neurotransmitter release can still occur.

This manuscript provides the most comprehensive study to date on the effects of mutant Syntaxin variants to assess their effects on synaptic transmission, -efficacy, and a lot of other synaptic determinants. The manuscript contains an extensive set of experiments of high-quality, which are clearly described and presented in the text and figures. All conclusions are justified and built on solid evidence. The main limitations of the current study are the strongly reduced expression levels of some mutants, an issue the authors also raise, which limit some of the interpretations, and the effects of N-peptide mutants which are underdeveloped. However, taken together, this study is of great interest to the synapse field and has all the ingredients to provide the final verdict on some controversial issues.

Major points:

1. The authors attempt to address the reduced protein expression by titrating down WT levels by using only 1/12 of the viral load. Strikingly, reducing WT STX1A levels by almost 6-fold had no effect on synaptic transmission. This raises important questions. At which point does STX1A become rate-limiting? And how do these protein levels compare to endogenous STXA levels? The authors should show the expression levels of their mutants relative to endogenous STX levels. Furthermore, a previous paper from the same group concluded that titrating Synatxin1 did impact synaptic transmission (Arancillo et al., J Neurosci 2013). The authors should discuss how their expression levels and findings compare to the findings in their previous paper.

2. While the effect of the N-peptide on most release parameters is absent/small, a robust effect was observed on short-term plasticity: up to70% larger EPSC's at the end of a 10Hz train in N-peptide mutants. This substantial effect is undervalued by the authors. Regulation of STP could be an important function of this domain. The authors mention this effect only briefly in the abstract, but should discuss this biological relevance in the Discussion. Second, typical traces of trains in Figure 6 should be of higher resolution and zoomed in more to critically evaluate. For instance, the standing (asynchronous) current that usually develops during 10Hz trains in autapses seems to be absent. Why is this the case? Since N-peptide mutations do not change PPR, the most logical explanation of the effect during 10Hz trains is a difference in the accumulation of global Ca during the train. This is plausible given the (disputed) interaction of syntaxin with calcium channels. To test this the experiments should be repeated +/- EGTA-AM. In this same experiment, the author can examine the potential effect on Ca-dependent replenishment or increase in fusogenicity (measuring RRP size and fusogenicity before and after 10Hz trains).*Reviewer #2 (Recommendations for the authors):*

This manuscript by Vardar, Rosenmund et al. provides a systematic analysis of syntaxin-1 function in exocytosis. It probes the major molecular roles of syntaxin through mutational analyses of three of its key features: the N-terminal peptide interaction with Munc18 (using short deletions), the role of the Habc domain (using a larger deletion), and the role of a confirmational switch (using the established LE open mutation). It probes the synaptic functions of these features either on their own or in combination with one another in rescue experiments in cultured neurons of syntaxin-1A and 1B null mutants, assessing synaptic transmission and its constituents, synaptic ultrastructure and neuronal survival.

The main finding is that the N-terminal peptide interaction with Munc18 is dispensable, and its role is only revealed when combined with the LE open mutation. Another finding is that syntaxin function is quite resilient, as even combining mutations (for example LE open and N-peptide deletion) does not fully disrupt synaptic transmission. In general, the finding that the N-peptide deletion is dispensable is important as it differs from previous studies that have used different manipulations. A key strength is the systematic comparison of the various mutants alone and in combination upon syntaxin-1 ablation and the testing of a series of different N-terminal deletions. The authors provide a balanced discussion of their and previous results, how they relate to one another, and how discrepancies could be reconciled.

Weaknesses in the current methodology include that the manuscript does not distinguish well between effects on cell health, effects on synapse numbers, and effects on synaptic properties. Furthermore, some of the data presentation relies on assumptions in the field rather than directly relating to the experimental approach. For example, the data describing the LE mutant and the N-peptide deletion assume that these mutations together essentially reveal a "complete loss-of-Munc18-binding" phenotype, but this may not necessarily be that case.*Reviewer #3 (Recommendations for the authors):*

In this manuscript, Vardar et al., investigated the role of different syntaxin I (STX1) domains in neurotransmitter release using a STX1-null mouse model system and exogenous reintroduction of STX1A mutants, the latter lacking either the N-peptide or the Habc domain of STX I. In addition, the STX1 mutant, which is present in the ‚open' conformation (LE open mutant) was examined with or without deletion of the N-peptide. The results show that the Habc domain is absolutely necessary for the stability or expression of STX1 and thus for neurotransmitter release. Moreover, it became clear, contrary to earlier work, that the N-peptide is dispensable for synaptic transmission, but assumes a regulatory role in controlling ca^2+^ sensitivity of vesicular release and generally in vesicle fusion when syntaxin is in the 'open' conformation. The manuscript provides a very comprehensive structure-function analysis using a wide variety of high-resolution techniques (e.g. HPF-EM, synaptic ca^2+^ imaging). Remarkably, perturbation of the STXI-Munc18 interaction by interfering with the canonical interaction modes has surprisingly little effect on synaptic transmission. Several results in this present manuscript correct observations of previous studies that used less quantitative and systematic approaches in the attempt to unravel the molecular mechanisms of syntaxin-Munc18 interaction in ca^2+^-driven exocytosis.

In general, the combined set of data is particularly valuable and allows new insights into the complexity of neuronal transmission and its underlying components.

---

## [Author Response]

Essential Revisions (for the authors):1. The authors attempt to address the reduced protein expression by titrating down WT levels by using only 1/12 of the viral load. Strikingly, reducing WT STX1A levels by almost 6-fold had no effect on synaptic transmission. This raises important questions. At which point does STX1A become rate-limiting? And how do these protein levels compare to endogenous STXA levels? The authors should show the expression levels of their mutants relative to endogenous STX levels. Furthermore, a previous paper from the same group concluded that titrating Synatxin1 did impact synaptic transmission (Arancillo et al., J Neurosci 2013). The authors should discuss how their expression levels and findings compare to the findings in their previous paper.

The aim of our down-titration experiments was not to determine which level of exogenous STX1A expression limits the neurotransmitter release, but we rather aimed to test whether the relative reduction in EPSC and in other release parameters in STX1A^LEOpen + ΔΝ^ neurons compared to that of STX1A^LEOpen^ neurons was due to the relative reduction in their STX1A levels. Therefore, we have reduced the number of viral particles used for 35 mm well from ~400.000 (1X) to ~35.000 for STX1A^WT^ and to ~ 65.000 for STX1A^LEOpen^ and successfully reached the levels of STX1A^LEOpen + ΔΝ2-28^. At this point, we did not observe any influence of expression levels on the release parameters neither by using STX1A^WT^ nor STX1A^LEOpen^. Therefore, we suggested that the reduction in the release parameters due to the N-peptide deletions on STX1A^LEOpen^ background is not due to its relatively lower expression levels but rather due to a functional deficit.

However, we thank to the reviewers for their suggestion to compare the endogenous STX1A levels to the exogenous STX1A levels used in this study to solve a possible discrepancy between this study and our previous study (Arancillo et al., 2013). It is important to note that our mouse line has the genotype of STX1B^FL/FL^; STX1A^-/-^. We do not currently hold a mouse line with the genotype of STX1B^FL/FL^; STX1A^+/-^ for breeding. Therefore, we cannot compare the endogenous and exogenous expression levels of STX1A using littermates. Because of that, we analyzed the endogenous STX1A levels in WT neurons in comparison to the lentiviral expression obtained with the help of 1X volume of lentiviral particles encoding STX1A in STX1-null neurons. However, we cultured both WT and the transgenic neurons on the same batch of astrocyte cultures and fixed them with PFA at the same DIV. We have observed that 1X volume of viral particles led to ~3 fold higher expression of STX1A compared to the endogenous levels (Figure 4 – Supplement 3). This shows that even the lowest expressing construct STX1A^LEOpen+ΔN2-28^ reaches an expression level of ~60% of endogenous STX1A^WT^. This also suggests that with our down-titration experiment we have reached again 60% of endogenous levels. We have explained and discussed our new data in relation to our previous study (Arancillo et al., 2013) in lines 236-252.

It should be noted that the WT neurons also express STX1B. It is fair to interpret our data as such that our experiments do not fall within over expression studies, because the neurons used in this study do not contain STX1B. Therefore, a ~3 fold higher expression of STX1A in the absence of STX1B constitutes a total level of STX1 comparable to that in WT neurons. However, because the STX1A^LEOpen+ΔN2-28^ do show a 60% expression of STX1A, we have removed our statement that ‘neurotransmitter release precedes normal even when both STX1A and Munc18-1 show a severely low expression’. Now, we have included our new interpretation that our down-titration experiments are insufficient to reach the rate-limiting level as determined by our previous study and that 60% of STX1A in the absence of STX1B is inadequate to rescue Munc18-1 levels (lines 242-249).

2. While the effect of the N-peptide on most release parameters is absent/small, a robust effect was observed on short-term plasticity: up to70% larger EPSC's at the end of a 10Hz train in N-peptide mutants. This substantial effect is undervalued by the authors. Regulation of STP could be an important function of this domain. The authors mention this effect only briefly in the abstract, but should discuss this biological relevance in the Discussion. Second, typical traces of trains in Figure 6 should be of higher resolution and zoomed in more to critically evaluate. For instance, the standing (asynchronous) current that usually develops during 10Hz trains in autapses seems to be absent. Why is this the case? Since N-peptide mutations do not change PPR, the most logical explanation of the effect during 10Hz trains is a difference in the accumulation of global Ca during the train. This is plausible given the (disputed) interaction of syntaxin with calcium channels. Ideally, the experiments should be repeated +/- EGTA-AM. In this same experiment, the author can examine the potential effect on Ca-dependent replenishment or increase in fusogenicity (measuring RRP size and fusogenicity before and after 10Hz trains). At the minimum, the observed effect should be more pointed out more clearly in the text.

Firstly, we thank to the reviewers for pointing out the importance of our finding of STX1A’s N-peptide’s regulation of STP. Now, we have discussed this issue also in our discussion with a greater emphasis (lines 520-538).

Secondly, we have now included zoomed-in example traces of absolute EPSCs for the first and last 5 stimuli for all the groups (Figure 6 – Supplement 2) for a better inspection, as suggested by the reviewers. It can also be observed in those zoomed-in traces that 10 Hz stimulation of autaptic neurons is not adequate for production of a measurable standing current. Usually, the currents elicited by 1 AP reach back the baseline in less than 100 ms, which is the time-interval for 10 Hz stimulation. Additionally, the decay-time of the EPSC affects the time required for the current to be back at baseline level and in autaptic neurons the bigger EPSCs can lead to longer decay-times and thus can contribute to the standing current (in-house information based on our previous experiments using low concentration of AMPA antagonists to produce smaller EPSCs). A reason for the effect of EPSC size on the decay-time and thus the standing current is the spill-over of glutamate because the autaptic neurons are not a fully-isolated system on the contrary of slice cultures. We understand that the reviewers would like to know the effect of N-peptide deletions and the LE_Open_ mutation on asynchronous release. However, more elaborate experiments such as the usage of Strontium instead of Calcium in the extracellular solution (Friedmann and Regehr, 1999, Biophys J) would be required.

Finally, we would like to raise some of our concerns regarding the usage of EGTA-AM for high-frequency stimulation. First of all, EGTA-AM experiments require pre-incubation of the neurons in EGTA-AM before electrophysiological recordings and thus leads to poorly controlled concentration of this ca^2+^-buffer in the neurons which ultimately may contaminate the results of high-frequency stimulation. Whereas small differences in the intracellular EGTA-AM concentration among individual neurons of one group might be negligible when the differences in the ca^2+^-sensitivity of the vesicular release and the RRP replenishment between two groups are high, they might greatly influence the interpretation of the data when those differences between two groups are small as in our study. Furthermore, increasing ca^2+^-buffering leads to complex changes in ca^2+^-transients during the trains of APs. During the train, ca^2+^-buffer saturates and removal of ca^2+^ through plasmalemmal pumps gets impaired leading to elevated intracellular ca^2+^-levels due to the extended time of the removal of ca^2+^ through the pumps at the end of the train. This further complicates the interpretation of the 10 Hz data using EGTA-AM particularly when the purpose of the study is the dissection of the mechanisms leading to small changes in STP and RRP replenishment. Thus, we prefer not to utilize EGTA-AM to examine the potential effect on ca^2+^-dependent replenishment or increase in fusogenicity. We would like to keep the focus of our study in that N-peptide is not indispensable for neurotransmitter release but rather regulates the STP and ca^2+^-sensitivity of the vesicular release.

However, we have discussed the N-peptides’s role in STP more with with a greater emphasis (lines 520-538) as mentioned before.

3. The main hypothesis put forward for reduced Ca-sensitivity and impaired RRP recovery in N-peptide mutants is an impairment in SNARE complex formation. How does this reconcile with up to 70% larger EPSC's during 10 Hz trains? Please comment and discuss.

We thank to the reviewers for raising this issue. We have calculated the net depression of EPSCs during 10 Hz trains for N-peptide deletions and saw that the responses recorded from STX1A^WT^ neurons depress down to 60% of the initial EPSC size and EPSCs recorded from N-peptide mutants depress down to 80-90%. We also have plotted the absolute EPSCs and observed that the EPSCs recorded from STX1A^ΔN2-19^ and STΧ1A^ΔN2-28^ did not become larger but inclined to remain smaller compared to those recorded from STX1A^WT^ neurons (Figure 6 – Supplement 1). However, because STX1A^ΔN2-9^ had initial EPSC comparable to that of STX1A^WT^ and because those neurons showed short-term depression of only 20%, the absolute EPSCs became ~30 % larger compared to that of STX1A^WT^. We have now discussed this issue in the light of their capability of SNARE complex formation and tuned down our interpretation (lines 520-529).

4. The data in figure 6 on the interaction between the two M18 binding modes of Stx1 are intriguing. However, different effects of N-peptide mutations in the WT and LE_Open background are difficult to reconcile, and not further addressed in the discussion. It is therefore difficult to interpret these data in the light of the N-peptide function. Please comment and discuss.

We have already discussed this issue in our discussion (lines 426-441, especially lines 437-441). In brief, the facilitation of the SNARE complex formation by N-peptide and the tightening of the Munc18-1 bound closed conformation by the absence of N-peptide are only minor (Burkhardt, 2008; Colbert, 2013). Furthermore, N-peptide’s effect might become negligible in a membranous environment when STX1’s TMR is present (Dawidowski and Cafiso, 2013).

We also propose that N-peptide regulates ca^2+^-sensitivity of the vesicular release, which is a common phenotype of N-peptide deletions on the WT and LE_Open_ background. However, we rather remain conservative for that issue and only speculate about this ‘newly discovered’ function of N-peptide.

5. In Figure 6F the N-peptide mutations in Stx1 LE_Open slow down depression during the first few stimulations, but have no effect on STP at the end of the train. This is opposite from the effect found at the WT background, where these mutations affect STP at the end of the train (Figure 6A). How these seemingly different results can be reconciled should be addressed in the discussion.

STP behavior upon a high-frequency stimulus depends on the initial Pvr, fusogenicity, and the ca^2+^-sensitivity of the SVs. We show in our study that deletion of N-peptide does not influence the fusogenicity of the vesicles neither on STX1A^WT^ nor on STX1A^LEOpen^ background (Figure 6). However, LE_Open_ mutation increases the degree of the SV fusogenicity, which remains high also when N-peptide is deleted (Figure 6). Therefore, the STX1A^LEOpen+ΔΝ^ mutants show a high degree of depression upon 10 Hz stimulation. On the other hand, we also show that N-peptide deletion decreases the ca^2+^-sensitivity of the vesicular release both on STX1A^WT^ and on STX1A^LEOpen^ background (Figure 6). However, the decrease of the ca^2+^-sensitivity on STX1A^WT^ background is relatively minor compared to that on STX1A^LEOpen^ background. This is also evident in that the decrease of the ca^2+^-sensitivity on STX1A^WT^ is not enough to significantly reduce the initial Pvr and EPSC. Therefore, it is plausible that the effect of global ca^2+^-accumulation in the presynapse can be observed only in the late phase of the stimulus, when N-peptide is deleted on STX1A^WT^ background. However, STX1A^LEOpen+ΔΝ^ mutants also show a high degree of fusogenicity and that might nullify the effect of reduced ca^2+^-sensitivity at the end of the train of APs. Our interpretations regarding this issue can be found at lines 335-338, 348-350, and 473-491.

6. Given the twofold higher EPSC amplitude and Pvr for Stx1 LE_Open, this mutant will most likely end up with similar EPSC amplitudes at the end of the train as for the WT. However, the LE_Open Δ_N mutants will have lower EPSC amplitudes at the end of the train. This could lead to different conclusions about the role of the N-peptide, especially when combined with similar experiments with these mutants as suggested in point 2.

We already have shown the absolute values of EPSCs during a 10 Hz stimulus for STX1A^WT^ and for all the mutants (Figure 6 – Supplement 1). Whereas STX1A^∆N2-19^ and STX1A^∆N2-28^ tended to remain to elicit smaller EPSCs throughout the high-frequency stimuli compared to those of STX1A^WT^ (Figure 6- supplement 1), EPSCs recorded from STX1A^∆N2-9^ neurons became ~30 % larger at the end of the stimulus compared to that of STX1A^WT^ (lines 290-294). On the other hand, the EPSCs recorded from STX1A^LEOpen^ neurons by 10 Hz stimulation tended to be initially larger, but they declined further compared to that of STX1A^WT^ neurons (Figure 6 —figure supplement 1). STX1A^LEOpen+∆N2-28^ mutant remained to produce smaller EPSCs throughout the high-frequency stimuli compared to those of both STX1A^WT^ and STX1A^LEOpen^ (Figure 6 —figure supplement 1) (lines 328-333).

These findings render the hypothesis likely that N-peptide deletions both on STX1A^WT^ and STX1A^LEOpen^ background reduce the efficacy of SNARE complex formation and therefore leads to smaller EPSCs throughout the train. The exception for that observation is that STX1A^ΔΝ2-9^ neurons show EPSCs initially comparable to that of STX1A^WT^, which does not reconcile with the hypothesis that the ca^2+^-sensitivity of SVs and STP is affected by N-peptide deletions due to an impairment in SNARE complex formation. We have now discussed this issue more in detail in lines (520-535).

7. It is interesting to see in Figure 6J that, for low external Ca, N-peptide mutations on a LE_open background rescue the paired-pulse ratio to WT levels, while for increasing external Ca levels this gradually reverts to the LE_open phenotype. This should be addressed in the discussion in the light of the different Ca sensitivities and fusogenicities for the different experimental groups.

We already have discussed this issue in lines 348-350 and suggested that the observation that N-peptide deletion leads to an altered behavior only in the initial phase of the 10 Hz stimuli – when STX1A’s open conformation is facilitated – is consistent with the reduced ca^2+^-sensitivity (Figure 6I) but unaltered fusogenicity (Figure 6F) of the vesicles. A cartoon illustrating a speculative model for the effects of fusogenicity, ca^2+^-channel-SV coupling, and ca^2+^-sensitivity of vesicular release can be found in Figure 8.

8. If reduced Ca-influx explains reduced Ca affinity in LE_open mutant (Figure 6I), how does this reconcile with the LE_open-deltaN mutant, which has significant less Ca influx compared to WT (Figure 7E), but no significant difference in Ca sensitivity? Some explanation should be added to the Discussion.

We would like to emphasize that our findings show that LE_Open_ mutant does not reduce but enhances the ca^2+^-affinity of the vesicular release, which is in line with previous reports (Gerber, et al., 2008; Acuna et al., 2014). In Figure 6I, the ca^2+^-dose response curve of STX1A^LEOpen^ deviates from that of STX1A^WT^ with a leftward shift showing that STX1A^LEOpen^ neurons release a bigger fraction of their release capacity at lower ca^2+^-concentration, thus they have a higher ca^2+^-sensitivity.

ca^2+^-sensitivity of the vesicular release is a convoluted function of SV fusogenicity and ca^2+^-channel–SV distance coupling. Our sub-saturating sucrose application shows that STX1A^LEOpen^ leads to a higher fusogenicity, which contributes to the higher apparent ca^2+^-sensitivity. However, we cannot conclude from our experiments that the vesicles are docked and primed at a closer distance to ca^2+^-channels, which should be further investigated. Therefore, we argue that even though ca^2+^-channel concentration might be reduced in STX1A^LEOpen^ neurons, which should be further corroborated with further experiments, ca^2+^-sensitivity might be increased with adequate ca^2+^-channel–SV distance and increased SV fusogenicity.

On the other hand, STX1A^LEOpen+∆N^ mutants might lead to an increase in the distance between docked/primed vesicles and the ca^2+^-channels, which is likely due to the loss of N-Peptide. Because those neurons retain high SV fusogenicity due to the ‘opening’ of STX1A, ca^2+^-sensitivity of the SV fusion as a convoluted function of SV fusogenicity and ca^2+^-channel–SV distance coupling would approach back towards the WT-like level.

We have already had discussed this issue in our previous version of the manuscript (previous version 468-483). However, we incorporated a more elaborate discussion regarding the relationship between ca^2+^-influx, fusogenicity, and ca^2+^-sensitivity (lines 509-519) and we hope we could address your concern. Furthermore, a cartoon illustrating a speculative model for the effects of fusogenicity, ca^2+^-channel-SV coupling, ca^2+^-influx, and ca^2+^-sensitivity of vesicular release can be found in Figure 8.

9. The authors' interpretation that the effect of calcium influx relates to expression levels and survivability is somewhat challenged by the fact that the LE mutant has the strongest effect on calcium influx, while it is by far not the mutant with the lowest expression levels, nor is it experiencing a decrease in survivability. Please comment and discuss.

We haven’t directly related the expression levels of STX1A to the level of global ca^2+^-influx, but rather proposed it as a possible hypothesis (lines 359-363). Therefore, we measured ca^2+^-influx from neurons expressing STX1A^WT^ but at a level as reduced as STX1A^LEOpen+∆N2-28^ and found no difference. In this respect, we have interpreted our data that STX1A^LEOpen^ reduces ca^2+^-influx due to a functional impairment rather than due to low expression level (lines 363-372). We are sorry if there was a confusion.

10. Throughout the manuscript, the authors quantify cell numbers as a measure of survival. Some mutations, for example syntaxin knockout and the DeltaHabc mutant rescue, strongly affect cell survival. For the interpretation of the synaptic physiology, a more meaningful parameter would be synapse numbers, especially for evaluating the relationship between docking (measured per synapse profile) and sucrose RRP (measured per cell). It appears possible, if not likely, that synapses are lost before cells die, and in this case correlating RRP and docking may be strongly confounded. The authors may have the data in mass cultures already, and it should be possible to quantify synapse numbers (for example via Bassoon puncta densities) in these existing data. This should be accompanied by a discussion of the systems used (mass cultures for morphology and autaptic cultures for electrophysiology). If synapse numbers are changed, the RRP charge and # of docked vesicles should not be directly correlated (as done in Figure 5E), because RRP charge strongly depends on synapse numbers, while the number of docked vesicles per synapse does not. The discussion should be adjusted accordingly.

We thank to the reviewers for raising the point of a possible alteration of synapse number in STX1A^LEOpen^ and STX1A^LEOpen+∆N2-28^ neurons as a possible reason for the reduction in their RRP. To test this hypothesis, we analyzed our VGlut1 immunocytochemical images of autaptic neurons using the images which were shown before in the Figure 4 – Supplement 2. We have found no difference in the synapse number nor in the synapse area among STX1A^WT^, STX1A^LEOpen^, and STX1A^LEOpen+∆N2-28^ neurons. We have included our finding as Figure 5—figure supplement 1 and the numerical values can be found in the respective source data. Because we found no reduction in synapse number but in RRP, we have kept our initial comment for the correlation between the number of docked vesicles and the RRP charge. On the other hand, we noted that the vesicle docking is assessed in mass cultures whereas RRP charge is assessed in autaptic cultures. Our explanation of our finding of the synapse number can be found in lines 265-272.

11. The authors present the double mutation of opening syntaxin (LE mutant) and N-peptide deletion as "interruption of both Munc18-1 binding modes" (caption line 185, figure captions Figures 4 and 5). While we appreciate where the terminology for considering this a "null for Munc18 binding" comes from, it would be much better to stay closer to the experiments in the data presentation and use wording like "opening of syntaxin combined with N-peptide deletion" instead. As the authors discuss, Munc18-syntaxin interactions are complex, and most data suggest that Munc18 also binds to SNARE complexes (and open syntaxin) while they are being assembled. Opening syntaxin and deleting the N-peptide does unlikely fully abolish Munc18 binding, but biases binding of syntaxin to Munc18 alone towards binding of Munc18 to syntaxin as it is being incorporated into SNARE complexes. The authors should do justice to this point in figure captions, and should discuss it as such.

We are sorry for using a terminology which might be misleading for the readers. We already have commented in our discussion that Munc18-1 is probably still bound to STX1A and/or SNARE complexes even when N-Peptide is deleted in STX1’s open conformation (lines 416-425). We have corrected the caption line 186 and the figure captions of Figures 4 and 5 as the reviewers suggested.

12. The effect of the LE mutant of syntaxin has been characterized by several biophysical methods that show that it shifts the conformational equilibrium towards the open state, but it still adopts the closed state upon Munc18 binding as observed by SAXS (Colbert et al. 2013) and single molecule FRET (Wang et al., EMBO J. 36, 816-, 2017). Moreover, the LE mutant only partially compensates for deletion of Munc13, i.e., the rescue is rather limited as observed in autaptic neuronal cultures and by single molecule FRET experiments (Lai et al., Neuron 95, 591-, 2017). Moreover, the results in Wang et al., 2017 suggest that the LE mutant interferes with proper Munc13 function (via its interactions with syntaxin), which may be related to the observed phenotypes of this mutant. Please discuss the experiments presented here in the context of these biophysical findings.

We have already discussed that STX1A^LEOpen^ binds to Munc18-1 in closed conformation (lines 418-422). We now included the references Wang, EMBO, 2017 and Lai, Neuron, 2017 into our discussion of this point (line 420). We also included these references into our discussion of Munc18-1 and Munc13’s roles as a template for SNARE complex formation (line 444-445). Additionally, we discussed the possible impairment of Munc13-STX1 interaction when STX1 includes LE_Open_ mutation (lines 448-454).